# A molecular mechanism for the topographic alignment of convergent neural maps

Elise Savier[1], Stephen J Eglen[2,3], Amélie Bathélémy[1], Martine Perraut[1], Frank W Pfrieger[1], Greg Lemke[4], Michael Reber[1,3]*

[1]CNRS UPR3212 – Institute of Cellular and Integrative Neuroscience, University of Strasbourg, Strasbourg, France; [2]Department of Applied Mathematics and Theoretical Physics, Cambridge Computational Biology Institute, University of Cambridge, Cambridge, United Kingdom; [3]University of Strasbourg Institute of Advanced Study, Strasbourg, France; [4]Molecular Neurobiology Laboratory, The Salk Institute for Biological Studies, San Diego, United States

**Abstract** Sensory processing requires proper alignment of neural maps throughout the brain. In the superficial layers of the superior colliculus of the midbrain, converging projections from retinal ganglion cells and neurons in visual cortex must be aligned to form a visuotopic map, but the basic mechanisms mediating this alignment remain elusive. In a new mouse model, ectopic expression of ephrin-A3 (*Efna3*) in a subset of retinal ganglion cells, quantitatively altering the retinal EFNAs gradient, disrupts cortico-collicular map alignment onto the retino-collicular map, creating a visuotopic mismatch. Genetic inactivation of ectopic EFNA3 restores a wild-type cortico-collicular map. Theoretical analyses using a new mapping algorithm model both map formation and alignment, and recapitulate our experimental observations. The algorithm is based on an initial sensory map, the retino-collicular map, which carries intrinsic topographic information, the retinal EFNAs, to the superior colliculus. These EFNAs subsequently topographically align ingrowing visual cortical axons to the retino-collicular map.

*For correspondence: michael.
reber@inserm.fr

## Introduction

Brain function relies on the efficient processing of sensory information, which in turn requires the proper formation and interaction of multiple sensory maps of the world. The superior colliculus (SC) of the midbrain is a major hub for sensory processing, as it receives organized inputs from visual, auditory, and somatosensory modalities (*May, 2006*). The SC is a laminated structure controlling visuo-spatial orientation and attention (*May, 2006*; *Krauzlis et al., 2013*). As such, defective visual processing in the SC has been associated with psychiatric conditions (*Mathis et al., 2015*). Visual information reaches the superficial layers of the SC, which are innervated both by retinal ganglion cells (RGCs - the retino-collicular projection) and by layer V neurons of the primary visual cortex V1 (the cortico-collicular projection). During development in the mouse, the retino-collicular map forms during the first post-natal week, and is followed by the cortico-collicular map, which develops between P6 and P12 (*Triplett et al., 2009*). These visuotopic maps must be aligned to ensure efficient modulation of the SC's retinal response by V1 inputs (*Zhao et al., 2014*; *Liang et al., 2015*). It has been suggested that the formation of the visuotopy is a stochastic process instructed by a balanced contribution of molecular cues and correlated neuronal activity (*Chandrasekaran et al., 2005*; *Tsigankov and Koulakov, 2006*; *2010*; *Triplett et al., 2009*; *Ackman and Crair, 2014*;

*Owens et al., 2015*). However the basic principles and underlying molecular mechanisms governing the alignment of converging maps have not been fully described.

Potential candidates are gradients of Eph receptor tyrosine kinases (EPHS) and their membrane-bound ligands, the ephrins (EFNs), already known to control retino-collicular map formation. In the mouse, EPHA4/A5/A6 receptors are present on projecting RGCs in a low-nasal to high-temporal gradient. RGC axons are repelled when their EPHAs contact and are activated by collicular EFNA2/A3/A5, expressed in a low-rostral to high-caudal gradient in the SC (*Lemke and Reber, 2005*; *Cang and Feldheim, 2013*). Counter-gradients of ligands (EFNA2/A3/A5) and receptors (EPHA3/A4/A7) are also present in the RGCs and the SC respectively, but their role remains controversial (*Feldheim et al., 2010*; *Triplett and Feldheim, 2012*; *Suetterlin et al., 2012*; *Weth et al., 2014*). In V1, gradients of EPHA4/A7, running from high-lateral to low-medial have also been observed, and evidence from genetic analyses suggests their involvement in the development of cortico-collicular projections (*Cang et al., 2005*; *Wilks et al., 2010*). Moreover, the formation of the cortico-collicular map requires retinal input (*Khachab and Bruce, 1999*; *Triplett et al., 2009*), but again, the underlying molecular mechanisms remain elusive.

The complex expression patterns of *Epha/Efna* gradients in both projecting and target structures require the development of incisive in vivo approaches to address, in a selective and quantitative manner, the function of these gradients in specific neuronal projections. Here, we analyzed the role of retinal *Efna* gradients in visuotopic map formation in the SC. We generated a unique mouse model in which EFNA3 is ectopically expressed only in *Isl2*(+) RGCs, therefore quantitatively disrupting the EFNA gradients exclusively in the retina without affecting *Ephas/Efnas* expression in other visually-related structures. Surprisingly, *Isl2-ires-Efna3* knock-in mice (referred as Efna3 KI) exhibited normal retino-collicular/geniculate maps and normal ipsi/contra-lateral projections. In marked contrast, the formation of the cortico-collicular map was severely disrupted, leading to a mix of single and duplicated projections and generating a mismatch with the retino-collicular map. The causal role of EFNA3 ectopic expression was further confirmed when in vivo inactivation by co-expressed EPHA3 receptor in the same *Isl2*(+) RGCs restored a wild-type map. Theoretical modelling recapitulated our experimental observations in normal and aberrant conditions induced by EFNA3 ectopic expression, therefore validating the basic principle and mechanism of map alignment. This mechanism suggests that a leading sensory map carries positional information cues for further sensory projections, within the same modality, to align and adjust.

## Results

### Knock-in mice for *Efna3* ectopic expression in *Isl2*(+) RGCs show normal retino-collicular and retino-geniculate projections

To test the role of retinal EFNA ligands in visuotopic map formation, we generated knock-in mice in which a full length *Efna3* cDNA was inserted, flanked by an internal ribosome entry site (ires), into the 3'-UTR region of the *Islet-2* (*Isl2*) gene (*Figure 1—figure supplement 1*), similar to a previous approach (*Brown et al., 2000*). Immunohistochemical staining confirmed ectopic expression of EFNA3 in somata and proximal axons of *Isl2*(+) RGCs in postnatal day 1 (P1) and P8 *Efna3* homozygous knock-in (Efna3KI/KI) mice compared to wild-type (WT) littermates (*Figure 1A–F*) without affecting *Isl2* expression at P1 (*Figure 1G,H*). Efna3KI/KI mice present two intermixed sub-populations of RGCs: *Isl2*(−) cells, expressing wild-type levels of *Efna3*, and *Isl2*(+) cells expressing additional *Efna3* (*Figure 1E–F,I–N*). EFNA3 is observed on RGC axons in vitro (*Figure 1O,Q'*). Ectopic expression of EFNA in *Isl2*(+) RGCs did not induce perceptible changes in synaptic layers or retinal organization (*Figure 2A,B*). Quantitative transcript (mRNA) measurement on RGC cell bodies, performed and standardized as described previously (*Reber et al., 2004*; *Claudepierre et al., 2008*) confirmed a two-fold increase of *Efna3* mRNA and normal levels of *Efna2/a5* mRNA in nasal, central and temporal Efna3KI/KI mutant retinas when compared to WT littermates (*Figure 2C*). Together, these data confirm a two-fold ectopic expression of *Efna3* in *Isl2*(+) Efna3KI/KI RGCs, which generates an oscillating high-nasal to low-temporal gradient in *Efna*s expression in the Efna3 KI mouse, very similar to the oscillating *Epha*s gradient we previously described for *Isl2-Epha3* knock-in mice (*Reber et al., 2004*).

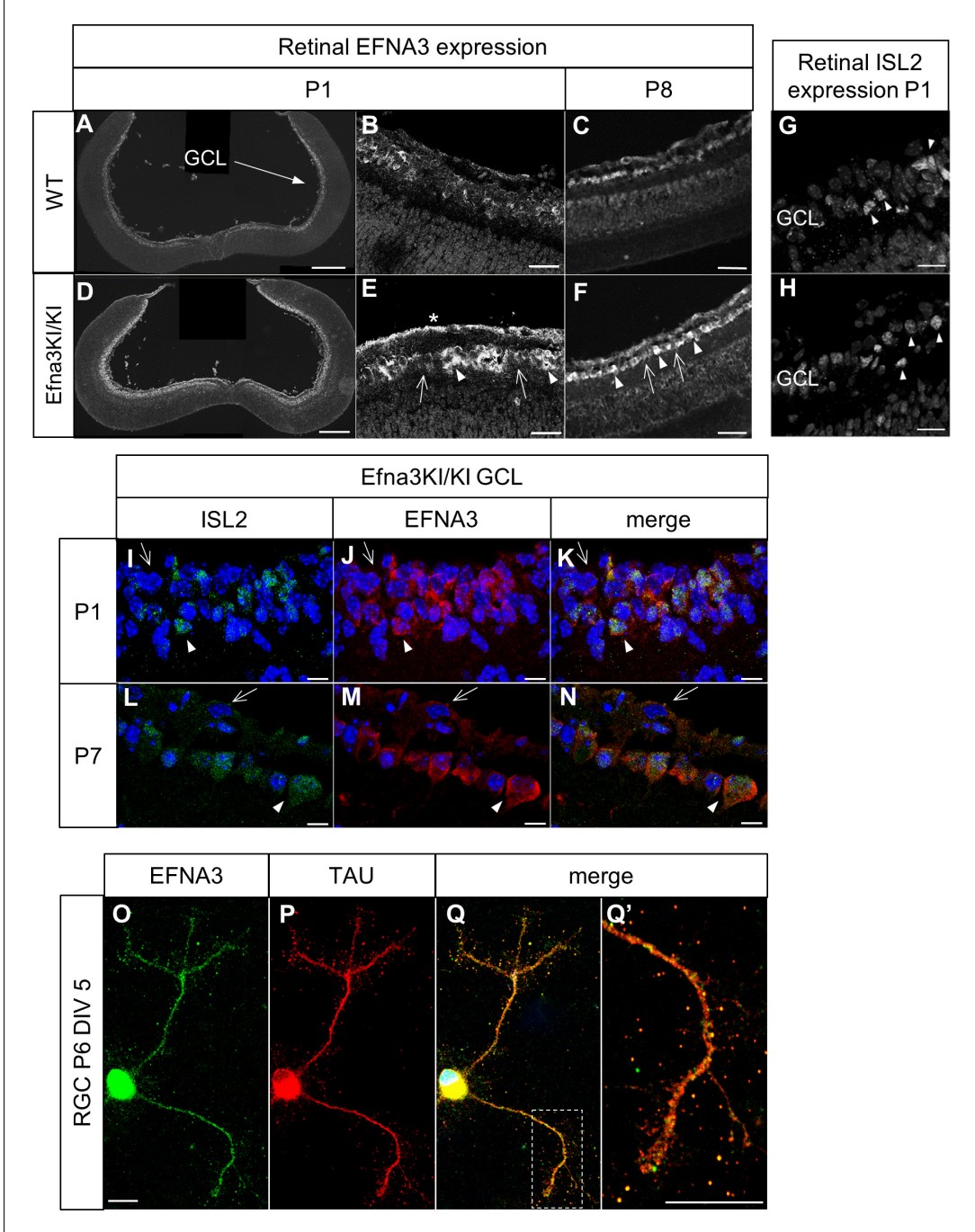

**Figure 1.** Validation of the Efna3 KI model I. (A–F) Immunochemical staining using anti-EFNA3 (LS-C6547, LS-Bio) in P1 (A, B) and P8 (C) WT (n = 3 animals) and P1 (D, E) and P8 (F) Efna3KI/KI (n = 5 animals) RGCs. In (E, F) arrows indicate EFNA3 WT expression level in RGCs, arrowheads indicate putative ectopic (knock-in) EFNA3 expression in RGCs. The asterisk highlights strong EFNA3 labelling in the RGC fiber layer. Scale bars represent 200 µm (A, D), 20 µm (B–F). (G–H) Immunostaining of ISL2 in P1 WT (n = 3 animals) (G) and Efna3KI/KI (n = 5 animals) (H) RGCs in the ganglion cell layer. Arrowheads indicate ISL2(+) RGCs. Scale bars represent 20 µm. (I–N) Immunostaining of ISL2 (I, L), EFNA3 (LS-C6547, LS-Bio) (J, M) and merged (K, N) in P1 and P7 Efna3KI/KI (n = 5 animals) RGCs. Arrowheads indicate ISL2(+) RGCs with high EFNA3. Arrows indicate ISL2(-) RGCs with wild-type EFNA3 level. Scale bars represent 10 µm. (O, Q') EFNA3 (LS-C6547, LS-Bio) (O, Q, Q') and axonal marker TAU (P, Q, Q') co-immunostaining on P6 RGC in culture (DIV 5) (n = 3 animals). (Q') shows an EFNA3/TAU co-labelled RGC axon at higher magnification. Scale bars represent 10 µm. GCL, ganglion cell layer; RGC, retinal ganglion cell; DIV, days in vitro; WT, wild-type.

The following figure supplement is available for figure 1:

*Figure 1 continued on next page*

*Figure 1 continued*

**Figure supplement 1.** Scheme of the knock-in strategy generating the *Isl2-ires-Efna3* mutant mouse.

To study retino-collicular map formation, we performed focal anterograde DiI labelling in the retina of P7 mice and analyzed the termination zones (TZs) in the SC at P8, as described previously (*Brown et al., 2000*; *Reber et al., 2004*; *Bevins et al., 2011*). For quantitative analysis, we measured the locations of the termination zones (TZs) along the rostral-caudal axis of the SC and the location of the focal DiI injections along the nasal-temporal axis of the retina (*Figure 3—figure*

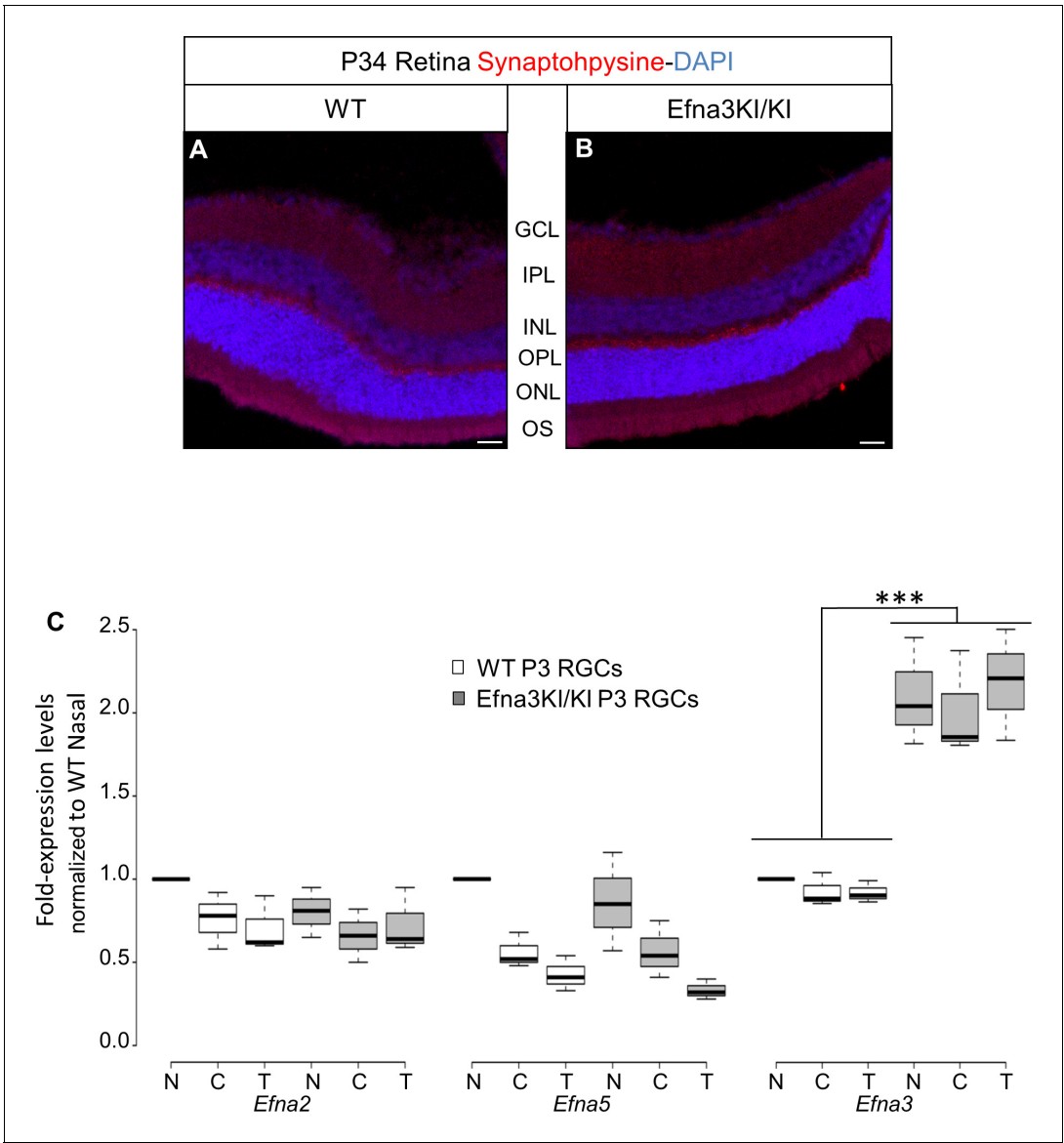

**Figure 2.** Validation of the Efna3 KI model II. (**A**, **B**) Synaptophysin (red) and DAPI (blue) staining in P34 WT (n = 4 animals) (**A**) and Efna3KI/KI (n = 4 animals) (**B**) retinas. Scale bars represent 50 μm. (**C**) *Efna* mRNAs relative to nasal WT quantified in acutely isolated RGCs from nasal (**N**), central (**C**) and temporal (**T**) retinas and normalized against *Gapdh* and *Hprt* in P3 WT and Efna3KI/KI (WT, n = 6 animals/12 retinas; Efna3KI/KI, n = 8 animals/16 retinas, variables are normally distributed, two-way ANOVA: *Efna2* genotype $F_{11,3}$ = 1.79, p = 0.11; *Efna5* genotype $F_{11,3}$ = 2.08, p = 0.07; *Efna3* genotype $F_{11,3}$ = 8.11, ***p<0.001). WT, wild-type; GCL, ganglion cell layer; IPL, inner plexiform layer; INL, inner nuclear layer; OPL, outer plexiform layer; ONL, outer nuclear layer; OS, outer and inner segments; RGCs, retinal ganglion cells.

*supplement 1*). Plotting these values in Cartesian coordinates, as described previously (*Brown et al., 2000*; *Reber et al., 2004*; *Bevins et al., 2011*), revealed normal linear retino-collicular maps in Efna3KI/KI and Efna3KI/+ mice, equivalent to those of WT littermates (*Figure 3A*). Detailed analysis showed that the size and the layering of the retinal TZs are similar between Efna3KI/KI and WT littermates (*Figure 3—figure supplement 2*). Retrograde labelling confirmed that axons from both *Isl2*(−) and *Isl2*(+) RGCs project to the SC (*Figure 3B–G*). Anterograde focal injections show normal retino-geniculate mapping (*Figure 3H–K*) and labelling by full-eye fills showed normal retino-collicular/geniculate eye-specific segregation (*Figure 3L–Q*) in Efna3KI/KI similarly to WT littermates. These results demonstrated that a two-fold increase of ectopic retinal *Efna3* expression in *Isl2*(+) RGCs does not disturb the formation of the retino-collicular/geniculate mapping nor eye-specific segregation.

## Cortico-collicular maps are duplicated in Isl2-Efna3KI mutants

To test whether retinal EFNA guidance cues influence the formation of the V1 cortico-collicular map, we traced cortico-collicular projections from V1 cortex by focal DiI injection in P14 mice and analyzed the location of the TZs in the SC at P15. Quantitative analyses using ImageJ revealed a remarkable duplication of the cortico-collicular map for single injections along the lateral-medial axis of the V1 in 47% of Efna3KI/KI (n = 9/19, *Figure 4A*, *Figure 4—figure supplement 1A*, *Figure 4—figure supplement 2A*) and 43% of Efna3KI/+ (n = 7/16, *Figure 4B*, *Figure 4—figure supplement 1B*) animals when compared to WT littermates (n = 9, *Figure 4C*, *Figure 4—figure supplement 1C*, *Figure 4—figure supplement 2B*) demonstrating heterogeneity of the phenotype in Efna3 KI mutants. If this heterogeneity is caused by genetic variations between animals, the same type of projections (either single or duplicated) should be observed in both colliculi of a given animal. This was not observed in 60% of Efna3KI/KI (n = 3/5) and 57% of Efna3KI/+ (n = 4/7) animals when we traced the cortico-collicular projections in both left and right colliculi (*Figure 4D,D'*). Therefore, genetic variation is unlikely to contribute to the map heterogeneity between animals of the same genotype. This heterogeneity is most likely the consequence of a stochastic process of map formation, as suggested previously (*Owens et al., 2015*). In Efna3KI/KI animals showing the same type of projections between colliculi, (40% of the mice) all of these projections were duplicated whereas in the Efna3KI/+ animals, the remaining 43% presented only single projections in both left and right colliculi, suggesting an effect of the level of *Efna3* ectopic expression onto cortico-collicular map duplication. As expected, all WT animals tested (n = 4) showed single projections in both colliculi (*Figure 4D*).

Next, we calculated the average distance of separation ($\Delta S_{exp}$) between the duplicated maps in Efna3 KI mutants as a percentage of the rostral-caudal axis of the SC (*Figure 4A,B*). This revealed a significant two-fold difference in map separation between Efna3KI/KI ($\Delta S_{exp}$ median = 13%, n = 9) and Efna3KI/+ ($\Delta S_{exp}$ median = 7%, n = 7) animals, which correlates with the presence of one or two copies of the *Isl2-ires-Efna3* allele (*Figure 4E*). Since *Isl2* is not expressed in the cortex (see *Figure 5B*), these results suggest that *Efna3* ectopic expression in *Isl2*(+) RGC axons/TZs destabilizes the stochastic process of cortico-collicular mapping, leading to map duplication in Efna3 KI animals. This in turn implies that V1 EphA-positive cortical axons sense varying EFNA3 levels on RGCs terminals by direct contact with these terminals in the SC (*Cang et al., 2005*) and anterograde labelling from retina and V1 cortex showed that both cortical and retinal axons terminals overlap in the superficial layers of the SC (*Figure 4F*) (*Phillips et al., 2011*; *Owens et al., 2015*). To further analyze the presence of EFNA3 of retinal origin in the SC, we performed EFNA3 immunohistochemistry coupled to retinal DiI injections on collicular sections. DiI labelled RGC axons were observed in the SC, as well as a strong collicular EFNA3 staining (*Figure 4—figure supplement 3A–F*, *Figure 5—figure supplement 1A–D'*). However, no EFNA3 staining on single DiI-labelled RGC axons could be detected in neither Efna3KI/KI nor WT littermates (*Figure 4—figure supplement 3A–F*). EFNA3 immunohistochemistry on P3 Efna3KI/KI and WT optic nerve (ON) longitudinal sections (proximal, middle and distal to the optic cup) showed the presence of EFNA3 on fibers within the proximal part of the ON in Efna3KI/KI and WT littermates (*Figure 4—figure supplements 4A,A', D,D'*) but not on the proximal part of Efna3−/− ON (*Figure 4—figure supplement 4H*). EFNA3 signal was also detected in the middle part of the ON in Efna3KI/KI but not in WT (*Figure 4—figure supplements 4B,B', E,E'*) and was nearly impossible to discern in the distal part of Efna3KI/KI ON (*Figure 4—figure supplements 4C,C', F,F'*), indicating that EFNA3 is present on RGC axons. This was

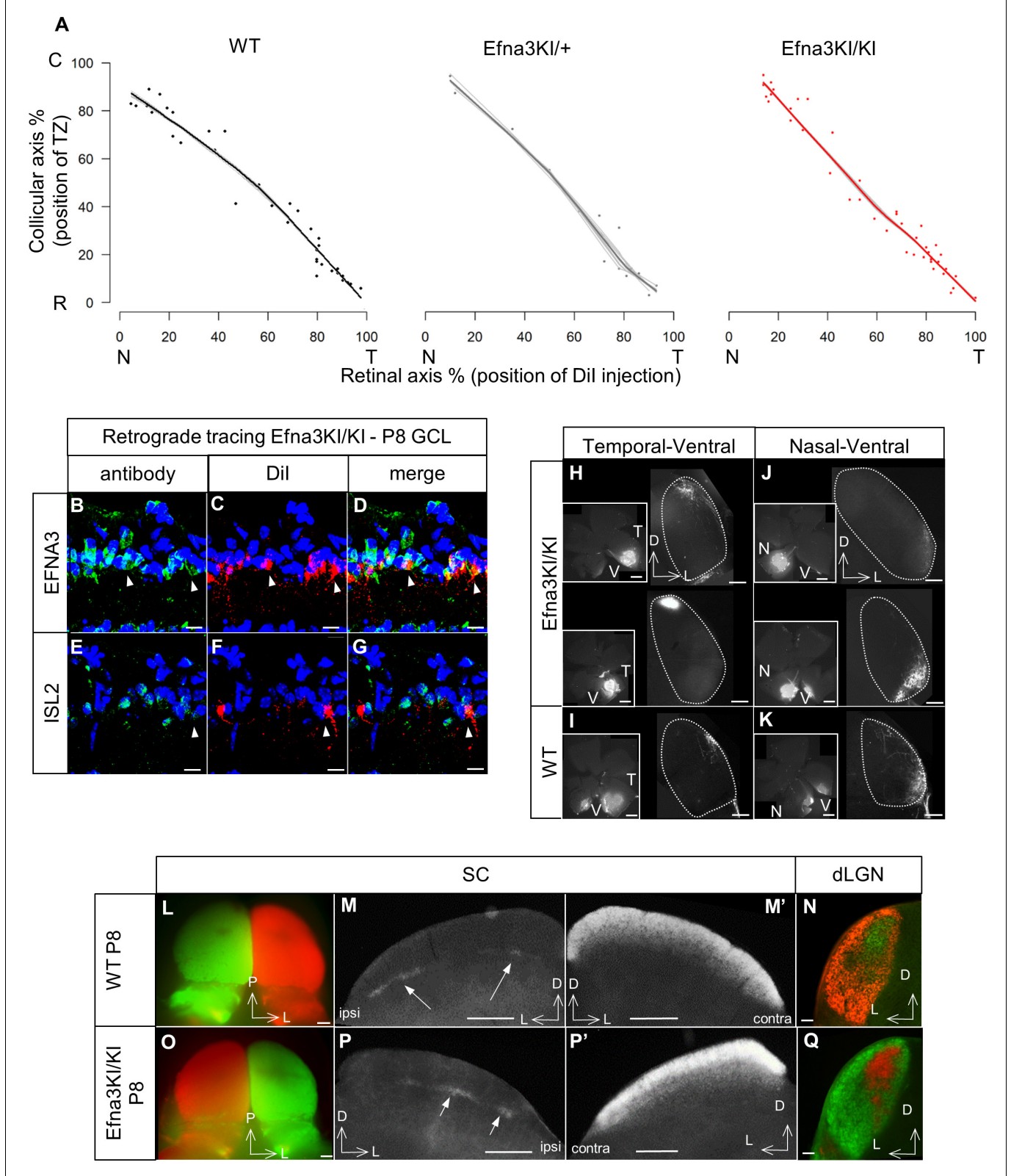

**Figure 3.** Retinal anterograde mapping and retrograde DiI labelling in P8 animals. (**A**) Retino-collicular maps generated by the Leave-One-Out method at P8 in WT (n = 38 animals), Efna3KI/+ (n = 15 animals) and Efna3KI/KI (n = 43 animals) mutants. Grey lines represent each map for each leave-one-out run. Each dot represents an individual mouse. (**B–G**) P8 DiI retrograde labelling (red) coupled to immuno-staining (green) of EFNA3 (**B–D**) and ISL2 (**E–G**) in Efna3KI/KI (n = 4 animals) mutant retinas. Arrowheads indicate ISL2(+) / high EFNA3 DiI labelled RGCs. Scale bars represent 10 μm. (**H–K**)

*Figure 3 continued on next page*

*Figure 3 continued*

Anterograde labelling after focal DiI injections in the retina showing retino-geniculate projections from the temporal-ventral (T–V) (H, I) and nasal-ventral (N–V) (J, K) retinal quadrants between Efna3KI/KI (H, J), (n = 4 animals) and WT (I), (K), (n = 4 animals) P8 littermates. Scale bars represent 400 µm. (L–Q) Anterograde labelling after full-eye injections showing eye-segregation in P8 WT (n = 3 animals) SCs (whole-mount top view: L); coronal sections: M, (M') and dLGN (N) and in Efna3KI/KI (n = 4 animals) SCs (whole-mount top view: O); coronal sections: P, (P') and dLGN (Q). Scale bars represent 250 µm (L–P'), 100 µm (N, Q). WT, wild-type; GCL, ganglion cell layer; T, temporal; N, nasal; R, rostral; C, caudal; dLGN, dorsal lateral geniculate nuclei; SC, superior colliculus.

The following figure supplements are available for figure 3:

**Figure supplement 1.** Anterograde DiI labelling showing the focal injection in the retina (white star in **A**) and the corresponding termination zones in the SC (**B**).

**Figure supplement 2.** Representative examples of collicular parasagittal sections showing labelled anterograde retinal termination zones in P8 Efna3KI/KI (**A**) and WT (**B**) littermates.

further confirmed by co-localization of EFNA3 with NF200 on fibers (*Figure 4—figure supplement 4I–J'* and *Figure 1B–F,O–Q'*). Retrograde labelling confirmed that the cortico-collicular projections originate from layer V neurons in V1 cortex in P14 Efna3KI/KI animals (*Figure 5A*). No bi-cistronic expression of *Isl2-ires-Efna3* in either SC or V1 cortex of WT and Efna3KI/KI animals could be detected (*Figure 5B*), ruling out any indirect effects of *Efna3* ectopic expression. Transcript analyses in colliculi and V1 cortices revealed similar levels of *Efna2/a3/a5* and *Epha4/a7* receptors in Efna3KI/KI compared to WT littermates at P7 (*Figure 5C*) excluding indirect effects due to local changes of *Ephas/Efnas* gene expression. Similar level of EFNA3 between Efna3KI/KI and WT littermates was further confirmed by immunostaining of EFNA3 (*Figure 5—figure supplement 1A–E*).

## Genetic cis-inactivation of ectopic EFNA3 expression in RGCs restores a wild-type cortico-collicular map

If the defective cortico-collicular maps in the Efna3 KI animals were solely due to *Efna3* ectopic expression in *Isl2*(+) RGCs, then inactivation of this ectopic expression should rescue the phenotype and restore a wild-type map. Previous work showed that co-expression of EFNA3 ligand and EPHA3 receptor in the same cell leads to their mutual inactivation through cis-masking (*Falivelli et al., 2013*). To accomplish this in vivo, we generated double heterozygous mice carrying *Epha3* on one allele of the *Isl2* gene and *Efna3* on the second allele (Epha3KI/Efna3KI). Immunohistochemical staining confirmed co-expression of EFNA3 and EPHA3 in acutely isolated double heterozygous RGCs (*Figure 6A*). Importantly, previous studies demonstrated that ectopic expression of the EPHA3 receptor in *Isl2*(+) RGCs in heterozygous Epha3KI/+ animals results in a duplicated retino-collicular map (*Figure 6—figure supplement 1*, *Brown et al., 2000*). Remarkably, co-expression of *Epha3* and *Efna3* in Epha3KI/Efna3KI double-mutant mice reverted the duplications, leading to normal retino- and cortico-collicular maps (*Figure 6B,C*). This double rescue of both maps indicates that cortico-collicular defects in Efna3 KI animals were caused by ectopic expression of EFNA3 in *Isl2*(+) RGCs and were restored by concomitant expression of EPHA3. The cis-inactivation mechanism was cross-validated by the presence of a normal retino-collicular map in the Efna3KI/Epha3KI double mutant mice, indicating EPHA3 inactivation (*Figure 6C*). To further evaluate any residual EFNA3 or EPHA3 signaling activity, we generated *Epha3KI/Efna3KIxEpha4* knock-in/out compound mutants. In these mice, decreasing the overall level of retinal EPHA receptors by eliminating EPHA4 expression would reveal subtle changes in retinal EPHA signaling strength (*Reber et al., 2004*; *Bevins et al., 2011*). According to the Relative Signaling model in the *Epha3KI/+::Epha4+/−* and *Epha3KI/+:: Epha4−/−* compound mutant animals analyzed previously (*Reber et al., 2004*) any residual EPHA3 signaling on map formation would generate duplicated retinal TZs, particularly in the caudal part of the SC where nasal RGCs axons, expressing low levels of EPHA receptors, project. Retinal DiI anterograde tracing revealed no retino-collicular duplications, even in the caudal-most pole of the SC, in *Epha3KI/Efna3KI::Epha4+/-* and *Epha3KI/Efna3KI::Epha4−/−* compound mutants, confirming inactivation of EPHA3 and EFNA3 in *Isl2*(+) RGCs (*Figure 6D,E*). Altogether, these results suggest that EPHA3 and EFNA3 ectopic expression in the same RGCs lead to their mutual inactivation and

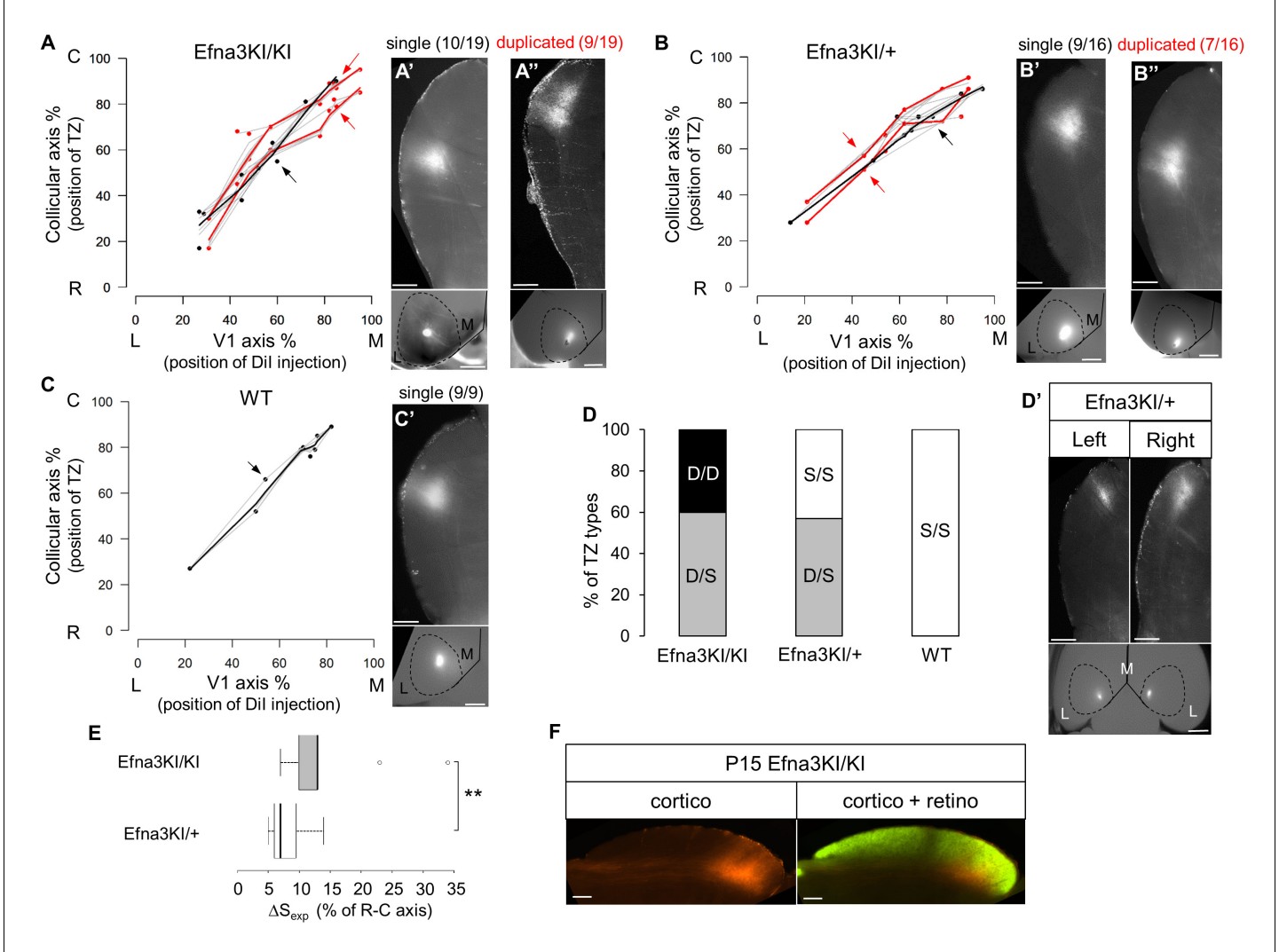

**Figure 4.** Anterograde cortico-collicular mapping in P14 animals. (A–C) Cortico-collicular maps generated by the Leave-One-Out method at P15 in Efna3KI/KI (n = 19 animals) (A), Efna3KI/+ (n = 16 animals) (B) and WT (n = 9 animals) (C) mutant littermates. Grey lines represent each map for each leave-one-out run for duplicated and single maps. Each dot represent an individual mouse. (A'–C') show examples of single (A', B', C') and duplicated (A'', B'') cortical TZs in collicular sagittal sections from different genotype together with the corresponding injections in V1 (lower panels in A'–C'). The corresponding coordinates are labelled by red arrows (duplicated) or a black arrow (single). Scale bars represent 400 µm (A'–C'), 1 mm (insets in A'–C'). (D–D') Percentage of heterogeneous duplicated/single (D/S) or homogeneous single/single (S/S) or duplicated/duplicated (D/D) projections in both colliculi of the same animal (D) (Efna3KI/KI, n = 5 animals; Efna3KI/+, n = 7 animals; WT, n = 4 animals). (D') shows an example of cortico-collicular TZs from left and right colliculi and corresponding V1 injections in a P15 Efna3KI/+ animal. Scale bars represent 400 µm (for colliculi), 1 mm (for V1). (E) Mean separation between duplicated maps measured experimentally ($\Delta S_{exp}$) in Efna3KI/KI (median = 13%, 1st quartile = 10%, third quartile = 13%, n = 9 animals) and Efna3KI/+ (median = 7%, 1st quartile = 6%, third quartile = 9.5 % n = 7 animals) (non-parametric Mann and Whitney test, **p=0.020). (F) Sagittal section of P15 Efna3KI/KI (n = 4 animals) DiI labelled cortical projection (left) and merged image of CTB-488 full-eye fill and cortical DiI projection (right). Scale bars represent 200 µm. L, lateral; M, medial; R, rostral; C, caudal; S, single; D, duplicated; TZ, termination zone; WT, wild-type.

The following figure supplements are available for figure 4:

**Figure supplement 1.** Representative examples of collicular parasagittal sections showing cortical terminations zones and the corresponding injections in V1 in Efna3KI/KI (A), Efna3KI/+ (B) and WT (C) P15 littermates.

**Figure supplement 2.** Collicular termination zones on parasagittal sections revealed after double DiI – DiD V1 anterograde injections in P15 Efna3KI/KI (A) and WT (B).

*Figure 4 continued on next page*

*Figure 4 continued*

**Figure supplement 3.** Immunostaining revealing EFNA3 expression (green) in P3 (**A, D**) and P8 (**B–F**) Efna3KI/KI and WT SC parasagittal sections after retinal anterograde DiI injection (red fibers).

**Figure supplement 4.** EFNA3 expression in Efna3 mutants and WT optic nerves.

confirm that cortico-collicular mapping duplications observed in Efna3 KI animals are the consequence of retinal EFNA3 ectopic expression.

## In silico modelling and theoretical analysis of cortico-collicular map alignment

Our results indicate that the level of EFNA3 on RGC projections innervating the SC influences the mapping and alignment of cortico-collicular projections. To simulate and explore the mechanism of map alignment, we created a 3-step map alignment model based on an algorithm originally developed to model retino-collicular mapping (*Tsigankov and Koulakov, 2006*; *2010*; *Owens et al., 2015*). Our version generates first the retino-collicular map based on retinal *Epha* receptors and collicular *Efna* ligands graded expression. The second step transposes the retinal *Efna* gradients onto the rostral-caudal axis of the SC according to the layout of the retino-collicular map generated in step one. In the third step, the cortico-collicular map is generated based on cortical *Epha* receptors expression and the transposed retinal *Efnas* in the SC. Each map is generated by a stochastic process based on balanced forces between repellent EPHA forward signaling and associating correlated neuronal activity (*Tsigankov and Koulakov, 2006*; *2010*; *Owens et al., 2015*) (see Experimental Procedures). To improve the validity of our model, we replaced the theoretical values of the retinal EPHA and EFNA gradients previously used (*Tsigankov and Koulakov, 2006*; *2010*; *Pfeiffenberger et al., 2006*; *Owens et al., 2015*) by our experimental quantification of retinal *Epha* mRNAs (*Reber et al., 2004*) ($R_A(x)^{retina}$) and *Efna* mRNAs ($L_A(x)^{retina}$) (*Figure 7A,B*) reasonably assuming, as made previously (*Reber et al., 2004*), that the measurement of mRNA differences translates linearly into differences in protein expression. $L_A(x)^{retina}$ best fit equations are:

$$L_{A5}(x)^{retina} = (1/0.56)^* \exp(-0.014x) \quad R^2 = 0.97 \tag{1}$$

$$L_{A2}(x)^{retina} = (1/0.54)^* \exp(-0.008x) \quad R^2 = 0.93 \tag{2}$$

$$L_{A3}(x)^{retina} = 0.44 \qquad\qquad R^2 = 0.97 \tag{3}$$

The total *Efna* gradients in the WT retina is:

$$
\begin{aligned}
L_A(x)^{retina\ WT} &= L_{A5}(x)^{retina} + L_{A2}(x)^{retina} + L_{A3}(x)^{retina} \\
L_A(x)^{retina\ WT} &= (1/0.56)^* \exp(-0.014x) + (1/0.54)^* \exp(-0.008x) + 0.44
\end{aligned}
\tag{4}
$$

The total *Efna* gradients in the mutant retinas are:

$$
\begin{aligned}
L_A(x)^{retina\ KI} &= L_A(x)^{retina\ WT} + \Delta L_{A3} \quad \text{with } \Delta L_{A3} = 0.22 \text{ in Efna3KI/+} \\
&\qquad\qquad\qquad\qquad \text{and } \Delta L_{A3} = 0.44 \text{ in Efna3KI/KI} \\
L_A(x)^{retinaKI/+} &= (1/0.56)^* \exp(-0.014x) + (1/0.54)^* \exp(-0.008x) + 0.44 + 0.22
\end{aligned}
\tag{5}
$$

$$L_A(x)^{retina\ KI/KI} = (1/0.56)^* \exp(-0.014x) + (1/0.54)^* \exp(-0.008x) + 0.44 + 0.44 \tag{6}$$

Equations were derived from semi-quantitative in situ hybridization as performed previously (*Reber et al., 2004*) (*Figure 7A,B*) and from our transcripts analyses measuring the relative expression levels of *Efnas* in acutely isolated RGCs (*Figure 2C*). We observed graded expression of *Efna2* and *Efna5* along the nasal-temporal axis of the retina, whereas *Efna3* is homogeneously expressed in WT animals (*Figure 7A,B*). Curve fitting analysis using MATLAB revealed the equation $L_A(x)^{retina\ WT}$

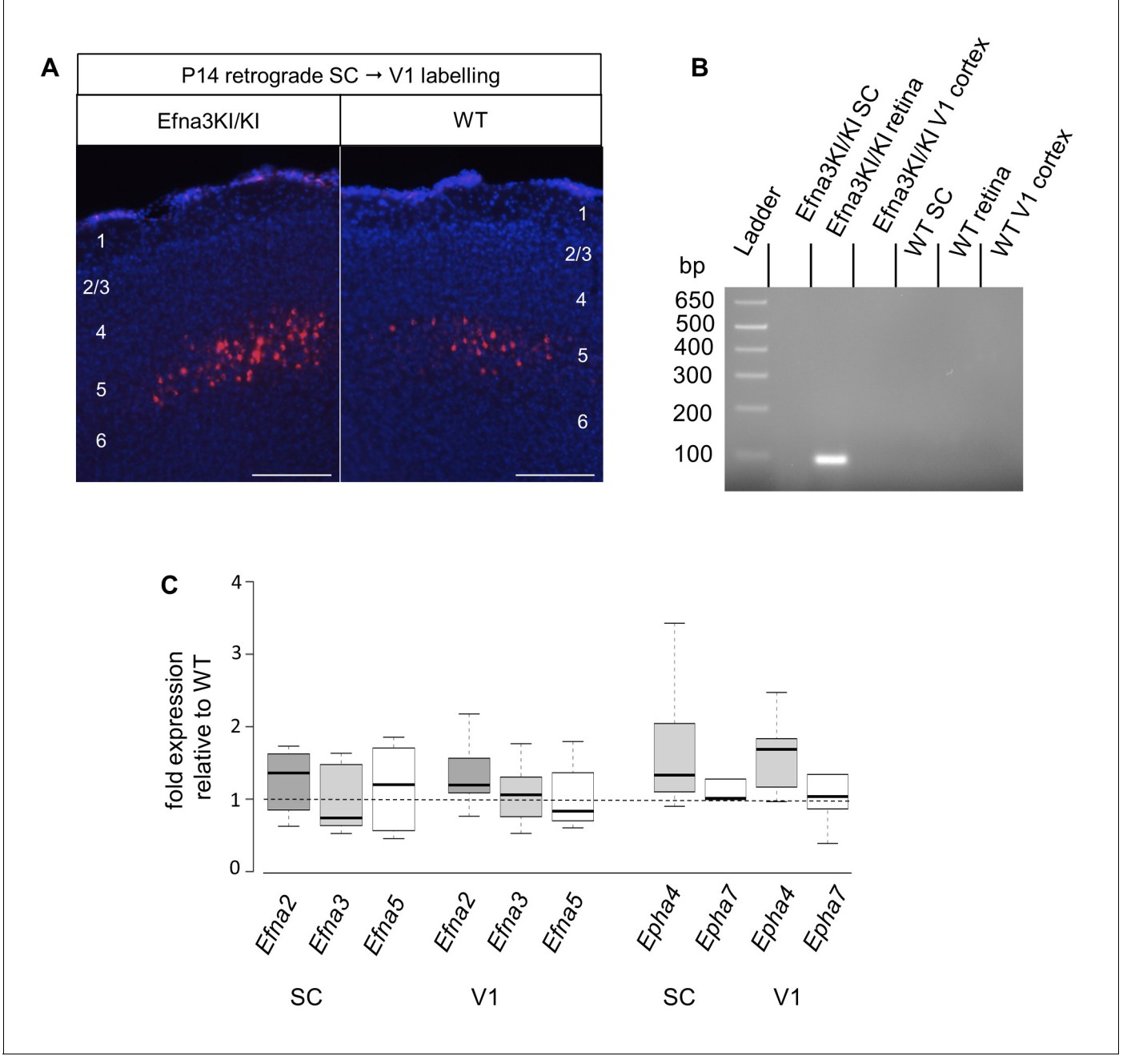

**Figure 5.** V1 retrograde labelling and *Ephas/Efnas* quantitative expression in V1 and SC. (**A**) Retrograde labelling by collicular DiI injection in P14 Efna3KI/KI (n = 4 animals) and WT (n = 3 animals) showing the localization of the projecting V1 cortical neurons in layer V (DiI, red; DAPI, bue). Scale bars represent 100 μm. (**B**) PCR amplification products from P7 RNA (35 cycles) in Efna3KI/KI (n = 3 animals per genotype). A single amplicon of 90 bp, corresponding to an amplified sequence located in the *ires* cassette of the targeted construct, is observed only in Efna3KI/KI retina. (**C**) Box-plot of *Efna2/a3/a5* and *Epha4/a7* mRNA fold-expression levels relative to wild-type (WT) at P7 in the superior colliculus and V1 cortex of Efna3KI/KI mutants (n = 4 animals for each genotype, two colliculi/V1 per animal, variables are normally distributed, one sample t-test: superior colliculus: *Efna2*, p = 0.17; *Efna3*, p = 0.48; *Efna5*, p = 0.50; *Epha4*, p = 0.09; *Epha7*, p = 0.26 – V1 cortex: *Efna2*, p = 0.09; *Efna3*, p = 0.61; *Efna5*, p = 0.84; *Epha4*, p = 0.07; *Epha7*, p = 0.58).

The following figure supplement is available for figure 5:

**Figure supplement 1.** EFNA3/DAPI immunohistochemical staining in P3 (**A – A'**, (**C, C'**) and P8 (**B – B'**), (**D**), (**D' E**) Efna3KI/KI (**A – B'**), WT (**C – D'**) and Efna3-/- (**E**) SCs sagittal sections.

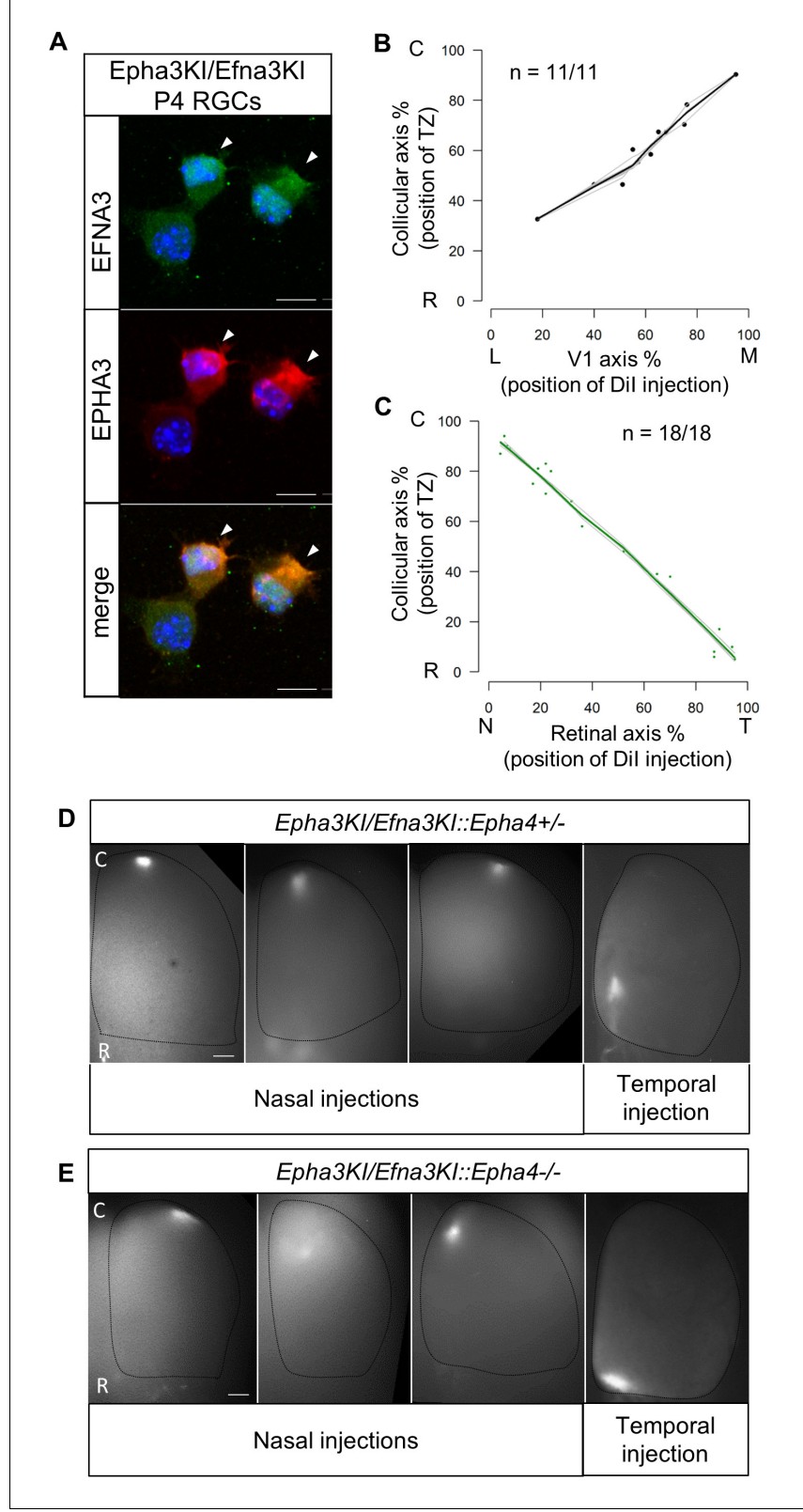

**Figure 6.** Retino- and cortico-collicular maps in *Epha3KI/Efna3KI* double heterozygous and compound *Epha3KI/Efna3KI::Epha4* knock-in/out mutants. (**A**) Co-immunostaining of EFNA3 (LS-C6547, LS-Bio; green) and EPHA3 (red) in P4 acutely isolated Epha3KI/Efna3KI (n = 8 animals/15 retinas) RGCs. Arrowheads indicate EPHA3/EFNA3 co-expression in the same RGCs. Scale bars represent 10 μm. (**B, C**) Cortico-collicular (n = 11 animals) (**B**) and retino-

*Figure 6 continued*
collicular (n = 18 animals) (C) maps generated by the Leave-One-Out method at P15 and P8 respectively in Epha3KI/Efna3KI double mutants. (D, E) Examples of retino-collicular projections for nasal and temporal retinal injections in P8 *Epha3KI/Efna3KI::Epha4+/-* (n = 6 animals) (D) and *Epha3KI/Efna3KI::Epha4-/-* (n = 6 animals) (E) mutants. Scale bars represent 200 μm. T, temporal; N, nasal; R, rostral; C, caudal; L, lateral; M, medial; RGCs, retinal ganglion cells.
The following figure supplement is available for figure 6:

**Figure supplement 1.** Retino-collicular map in P7 Epha3KI/+ mutants (n = 11 animals).

modelling WT retinal *Efna* ligands expression along the nasal-temporal axis (*Equations 1-4*). The two-fold increase of *Efna3* in Efna3KI/KI RGCs compared to WT (*Figure 2C*) was included into the model by adding a constant $\Delta L_{A3}$ to $L_A(x)^{\text{retina WT}}$ thus generating the $L_A(x)^{\text{retina KI}}$ alternating ectopic expression in the Efna3 KI retinas (*Equation 5, 6*). The 3-step map alignment model simulates the

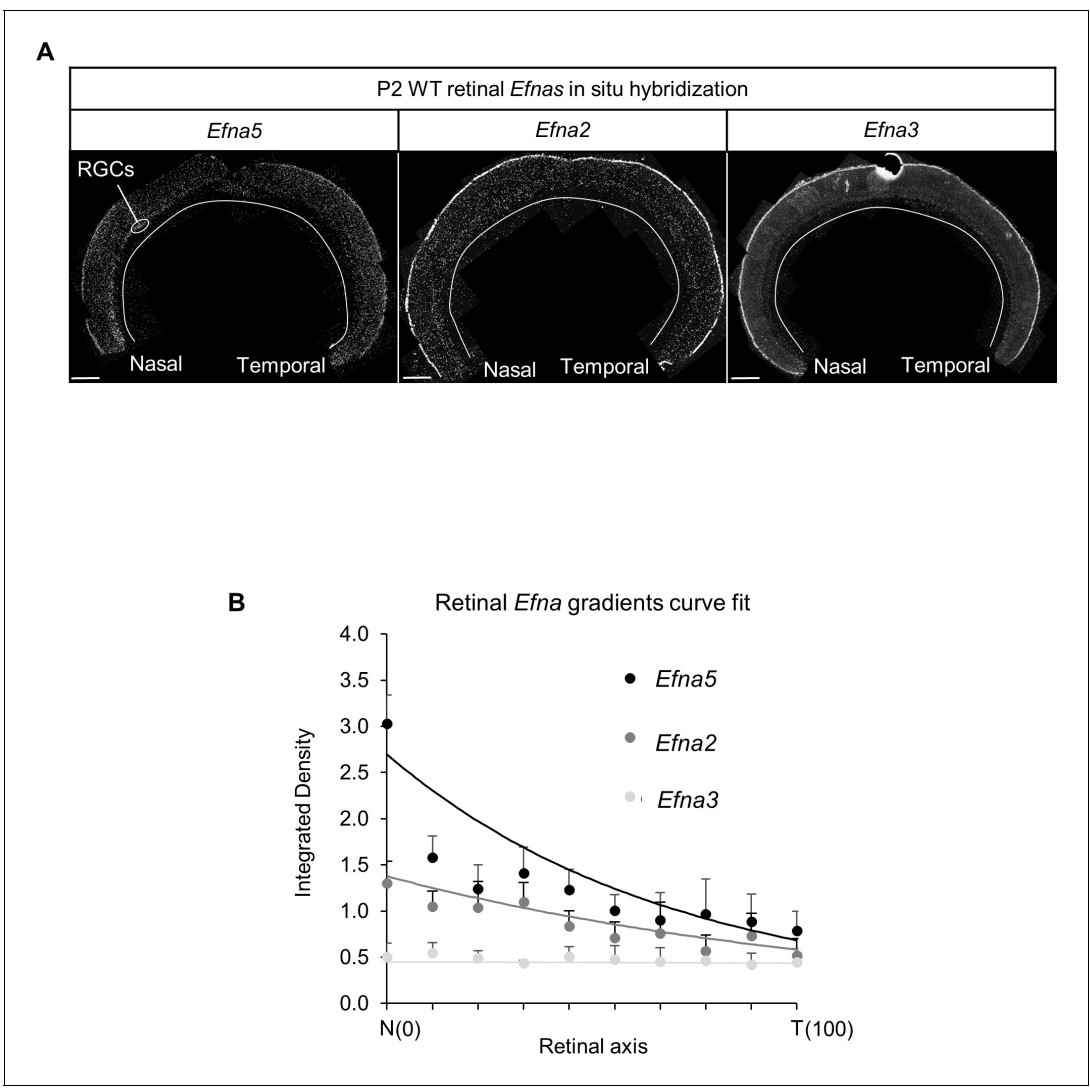

**Figure 7.** Retinal Efnas quantification. (**A**) Section of P2 mouse retina (n = 6 retinas from three animals from different litters) hybridized with *Efna5, Efna2* and *Efna3* probes. Quantification was performed for 10 segments of the RGC layer along the nasal-temporal axis. Scale bar represent 100 μm. (**B**) *Efna2/a3/a5* expression profiles (dots) fitted by the equations (lines) corresponding to *equations (1), (2) and (3)* respectively.

sequential mapping of 100 RGCs onto a 1D array of 100 SC neurons along the rostral-caudal axis, followed by 100 V1 cortical neurons innervating the SC. We made two assumptions: (1) endogenous collicular EFNAS are no longer active for incoming V1 axons as they have been engaged previously in RGCs axon guidance by binding to retinal EPHAS (forward signaling) which leads to the cleavage of their extracellular domains (*Janes et al., 2005*), (2) retinal EFNA3 alone cannot provide positional information in the SC as its expression is not graded in the RGCs. Consequently, a proportion of graded retinal EFNA2/A5 acts together with retinal EFNA3 to provide positional information in the SC. After $10^7$ iterations per run (n = 20 runs) for each genotype, stable and organized retino-collicular maps were formed (*Figure 8A,D,G*). Thereafter, a proportion of retinal *Efna* gradients were transposed in the SC (*Figure 8B,E,H*) following to the retinal projections layout and then the cortico-collicular maps are generated in a similar fashion (*Figure 8C,F,I*). To account for map heterogeneity observed experimentally, we further analyzed the maps generated using a linear regression (*Figure 8C,F,I*, red lines) and an exclusion parameter (EP) (*Figure 8F,I*, dashed grey lines), which corresponds to the variability of the WT single map (*Figure 8C*, dashed grey lines, $\sigma_{WT}$= 2.18 %) added to the genotype-specific average map separation ($\Delta S_{exp}$) calculated in Efna3 KI animals (*Figure 4E*, $EP_{KI/KI}$ = 15.18%; $EP_{KI/+}$ = 9.18 %; *Figure 8F,I*). The points, simulating the position of the cortico-collicular TZs along the rostral-caudal axis of the SC, located outside EP correspond to duplicated projections whereas the points located within EP correspond to single projections (*Figure 8F,I*). The percentages of duplicated projections generated by the model were similar to the percentages of experimentally measured duplications (*Figure 4A,B*) for both Efna3KI/KI and Efna3KI/+ animals (one sample t-test, *Efna3KI/KI*, p = 0.17; *Efna3KI/+*, p = 0.22; *Figure 8J*).

These results indicated that the 3-step map alignment model simulates both the retino- and cortico-collicular mapping and accurately recapitulates the normal and defective visual maps. It predicts the stochastic nature of the mapping abnormalities in Efna3KI/+ and Efna3KI/KI animals due to the *Efna3* ectopic expression in a subset of RGCs. Hence the model provides further evidence that retinal EFNA3 contributes to the alignment of the cortico-collicular map by providing positional information in the SC for ingrowing V1 axons carrying EPHAS.

## Discussion

EPHAS and EFNAS are present in opposed gradients within visual areas and in corresponding gradients between connected areas. These features preclude the identification of basic molecular mechanisms using full knock-out approaches. We therefore generated a unique and powerful mouse model in which gradients of EFNAS are *quantitatively* perturbed in the *RGCs only*. Together with theoretical modelling, we describe a molecular mechanism and associated principles governing the alignment of converging topographic neural maps in the brain.

### Retinal EFNA3 in cortico-collicular mapping

We showed that modestly elevated expression of EFNA3 exclusively in a subset of RGCs disturbed cortico-collicular map alignment in the SC, pointing to a mechanism where retinal EFNA3 provides positional information to ingrowing V1 cortico-collicular axons. Further confirmation came from the genetic inactivation of ectopic EFNA3 using co-expressed EPHA3 receptor. EPHA3/EFNA3 coexpression in the same cell, including RGCs, abolishes their trans-binding to EFNAS and EPHAS (*Connor et al., 1998*; *Hornberger et al., 1999*; *Menzel et al., 2001*; *Carvalho et al., 2006*; *Falivelli et al., 2013*; *Klein and Kania, 2014*). We therefore generated double heterozygous mutants, Efna3KI/Epha3KI. The power of this approach resides in the fact that each individual Efna3KI/+ and Epha3KI/+ mutant shows robust visuotopic map abnormalities (*Brown et al., 2000*; *Triplett et al., 2009*). The presence of WT retino- and cortico-collicular maps in the Efna3KI/Epha3KI double heterozygous mutants provides compelling evidence that both EFNA3 and EPHA3 were inactivated in *Isl2*(+) RGCs. Further evidence of EFNA3/EPHA3 inactivation came from the presence of normal retino-collicular maps in compound mutants *Efna3KI/Epha3KI::Epha4* knock-in/out mutants. Although we cannot rule out the occurrence of an altered expression/function of other yet unidentified molecules involved in mapping (e.g., semaphorins, L1) by ectopic EFNA3 expression, our results confirm the causal role of retinal EFNA3 ectopic expression on cortico-collicular alignment defects. They suggest that retinal projections play an instructive role in cortico-collicular map formation.

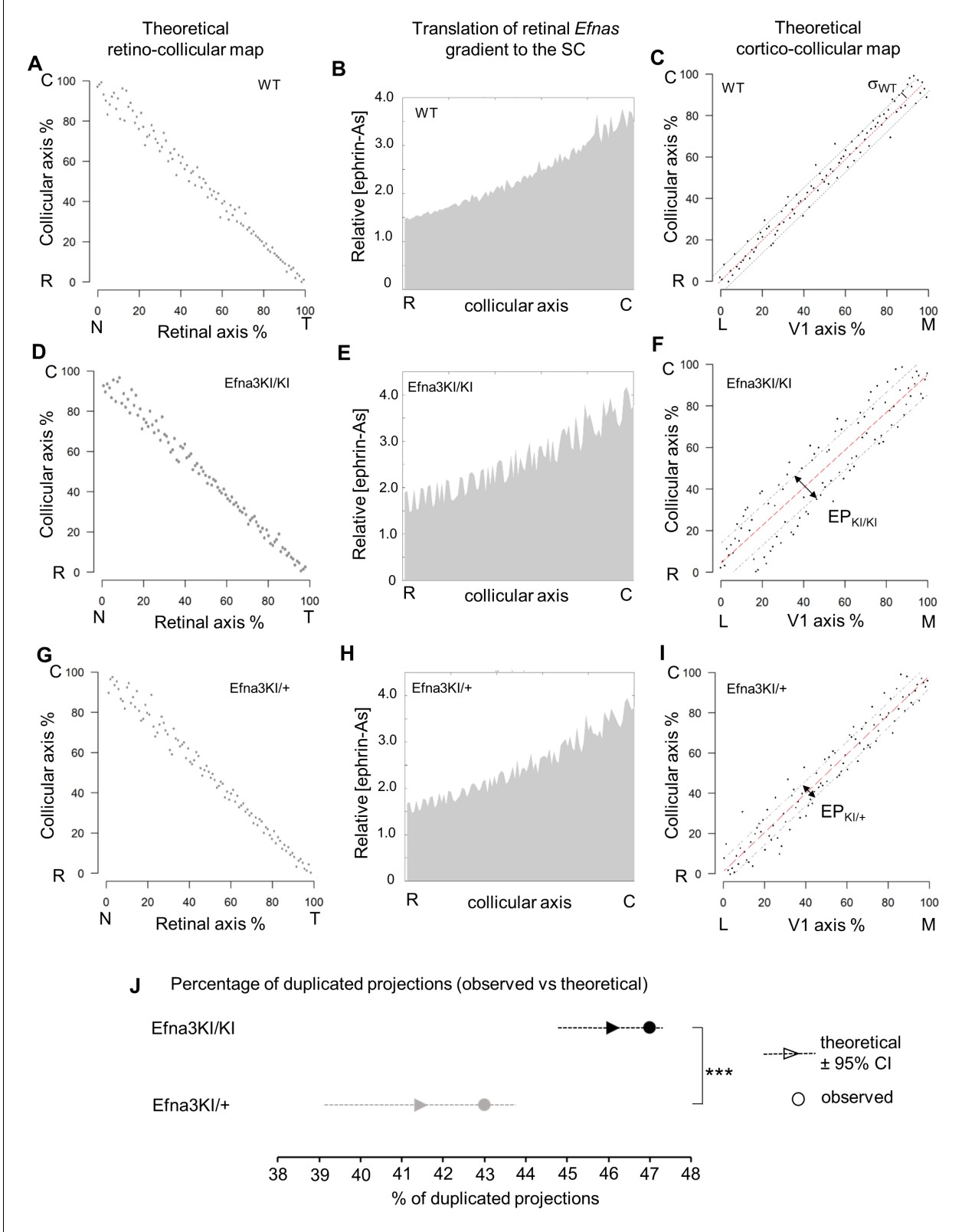

**Figure 8.** Theoretical analysis and modelling. (**A–I**) Retino- (**A, D, G**) and cortico- (**C, F, I**) collicular maps generated by the 3-step Map alignment model in WT (**A, C**), Efna3KI/KI (**D, F**) and Efna3KI/+ (**G, I**) after n = 20 runs and $10^7$ iteration/run. The parameters used for the retino-collicular modelling in the three genotypes are identical. Translated retinal Efn-As gradients (% of retinal expression: *Efna2* = 100%, *Efna3* = 100% and *Efna5* = 80%) into the SC in WT (**B**), Efna3KI/KI (**E**) and Efna3KI/+ (**H**). Red lines in (**C, F, I**) represent the linear regression. Variability of the WT map in (**C**) is calculated by $\sigma_{WT}$ =

*Figure 8 continued on next page*

*Figure 8 continued*

2.18% (n = 20 runs). Grey lines in (**F, I**) represent the exclusion parameter (EP) corresponding to EP = $\sigma_{WT} + \Delta S_{exp}$: $EP_{KI/KI}$ = 15.18%, $EP_{KI/+}$ = 9.18%. (**J**) Dot plot representation of the percentage of duplicated cortico-collicular projections (observed versus theoretical) for n = 20 runs in Efna3KI/KI (experimental = 47%, n = 9 animals; theoretical mean = 46.1%, ±95 %CI = 1.2%, one sample t-test, t = 1.42, 19 d.f., p = 0.17) and Efna3KI/+ (experimental = 43%, n = 7 animals; theoretical mean = 41.4%, ±95 %CI = 2.4%, variables are normally distributed, one sample t-test, t = 1.28, 19 d.f., p = 0.22) (theoretical Efna3KI/+ vs Efna3KI/KI, two-sample t-test, t = 3.4, 38 d.f., ***p = 0.0016). N, nasal; T, temporal; L, lateral; M, medial; R, rostral; C, caudal.

The heterogeneity of the cortico-collicular phenotype in the Efna3 KI mutants, revealed by a mix of single and duplicated projections, is in accordance with previous data showing a variable penetrance of the retinotopic mapping abnormalities in constitutive or conditional *Efnas* knock-outs (*Feldheim et al., 2000*; *Cang et al., 2008*; *Suetterlin and Drescher, 2014*; *Owens et al., 2015*). Such variable penetrance of the mutant phenotype can be explained by the stochastic nature of map formation driven by opposing forces resulting from EPHA signaling, which tends to separate neighboring RGCs through repulsion, and correlated neuronal activity, which tends to reinforce neighboring RGCs projections on adjacent target cells (*Cang et al., 2008*; *Triplett et al., 2009*; *Owens et al., 2015*). The general pattern of V1 collicular projections is consistent with the involvement of cortical EPHA receptor gradients (high-lateral to low-medial) (*Cang et al., 2005*) repelled by low-rostral to high-caudal EFNA gradients of retinal origin in the SC (forward signaling). In the Efna3 KI animals, retinal EFNA3 oscillation in the SC organizes neighbor-neighbor relationships of V1 and RGC TZs locally through repulsion inducing a small distance of map duplication. Our model is consistent with a retinal-matching model, suggesting that retinal inputs are required for proper cortico-collicular mapping (*Khachab and Bruce, 1999*; *Triplett et al., 2009*; *Cang and Feldheim, 2013*).

In our model, retinal inputs carry molecular cues, EFNA3 and likely other retinal EFNAs present at the level of the TZs to provide positional information for ingrowing V1 axons (*Figure 9*). EFNA3 could be detected on RGC axons by immunohistochemical staining on proximal and middle parts of the optic nerve but fell below detection limit in the distal part and in the SC. We cannot exclude that a proportion of collicular EFNAs also contributes to the cortico-collicular mapping process. Altogether, our data suggest that transposed retinal EFNAs in the SC instruct cortico-collicular map alignment and act together with correlated neuronal activity pattern shared between RGCs and V1 axons (*Triplett et al., 2009*; *Cang and Feldheim, 2013*). Triplett and collaborators (*Triplett et al., 2012*) suggested a gradient-matching model which posits that collicular EFNAs are required for the mapping of somatosensory inputs to the SC. These inputs behave similarly to the retino-collicular projections as they also require collicular EFNAs, although in different layers. In contrast, the requirement of SC-produced EFNAs for cortico-collicular mapping is unlikely. As mentioned by Triplett and colleagues, in this particular scenario, cortico-collicular projections in previously characterized Epha3KI/KI mutants would have led to a single TZ in the SC, leading to a mismatch between cortico- and retino-collicular maps. However, this was not observed (*Triplett et al., 2009*).

## Retinal EFNA3 in retino-collicular mapping

The presence of non-duplicated retino-collicular maps in both Efna3KI/+ and Efna3KI/KI mutants, as revealed by DiI tracing, suggests that retinal EFNA3 does not play a significant role in the formation of this map, consistent with previous work on Efna3-null mutants (*Pfeiffenberger et al., 2006*). Several hypotheses have been raised as to how retinal EFNAs may participate in retino-collicular map formation using in vitro, ex vivo and in vivo approaches in mouse and chick. For example, EFNAs on RGC axons are activated by collicular EPHAs (reverse signaling), leading to axon repulsion (*Rashid et al., 2005*; *Lim et al., 2008*; *Yoo et al., 2011*) or branch inhibition in the SC (*Yates et al., 2001*). In our mouse model, this mechanism would generate a segregation between *Isl2*(−) and *Isl2*(+) RGC axons in the SC, the latter being more repelled by collicular EPHAs. However, such a segregation did not occur, suggesting that retinal EFNA3 is not involved in such a repulsive mechanism by collicular EPHAs. In vitro and in vivo transfection in chick neurons suggested that retinal EFNAs bind to co-expressed retinal EPHAs in the same RGCs leading to inactivation/masking of the EPHA receptors rendering those axons less sensitive to EFNAs binding in the target tissue (*Connor et al., 1998*; *Hornberger et al., 1999*; *McLaughlin and O'Leary, 1999*; *Menzel et al., 2001*; *Yin et al., 2004*;

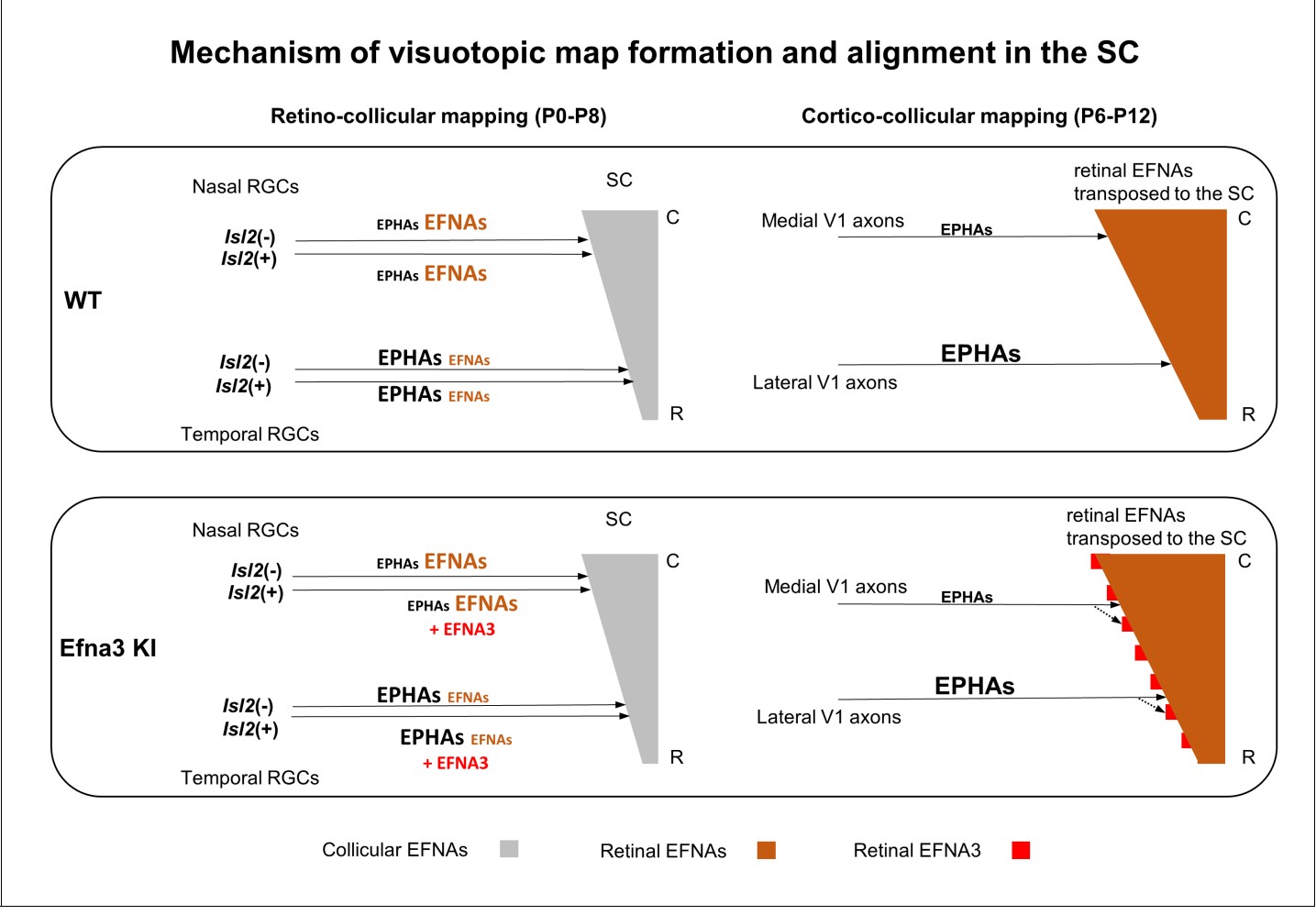

**Figure 9.** Schematic representation of the map alignment mechanism. RGCs axons in WT and Efna3 KI animals project to the SC during the first postnatal week and form the retino-collicular map through forward signaling activated by collicular EFNAs and fiber-fiber interactions (left side). In both WT and Efna3 KI animals, the retino-collicular map is single and coherent. In WT and Efna3 KI, each RGCs axon is loaded with a different concentration of retinal EFNAs (high-nasal, low-temporal) carried to the SC. Due to the coherence of the retino-collicular map, a smooth retinal EFNAs gradient is formed in the SC in WT, whereas in Efna3 KI animals, *Isl2*(+) RGC axons carry extra EFNA3 (in red) which creates an oscillatory retinal EFNAs gradient once transposed in the SC (right side). During cortico-collicular mapping, V1 axons carrying EPHA receptors are facing a smooth gradient of retinal EFNAs in the SC, leading to a single coherent map in WT. In Efna3 KI animals, V1 axons carrying EPHA receptors are facing an oscillatory gradient of retinal EFNAs (due to ectopic expression of EFNA3 in *Isl2*(+) RGCs) which leads to the duplication of the cortico-collicular map (43% in Efna3KI/+ and 47% in Efna3KI/KI) through local neighbor–neighbor relationships of V1/RGCs TZs via repulsion. C, caudal; R, rostral; WT, wild-type; TZ, termination zone; RGCs, retinal ganglion cells.

*Carvalho et al., 2006*). In our model, this mechanism would lead to a stronger inactivation of the EPHA4/A5/A6 receptors in the *Isl2*(+) RGCs expressing high levels of EFNA3, compared to RGCs with normal EFNA3 expression level. This would create two population of RGCs with different EPHA signaling strength, similarly to the EphA3KI mutants (*Brown et al., 2000*; *Reber et al., 2004*; *Triplett et al., 2009*; *Bevins et al., 2011*). In this context, according to the Relative Signaling model (*Brown et al., 2000*; *Reber et al., 2004*; *Bevins et al., 2011*), duplicated retino-collicular maps (partial or full) should be observed. Our results rather indicate that EFNA3 ectopic expression does not inactivate endogenous co-expressed EPHA4/A5/A6 receptors in RGCs. Nevertheless, we showed specific inactivation of EPHA3 by ectopic EFNA3, suggesting distinctive interactions in cis between EFNA/EPHA pairs as previously observed (*Yin et al., 2004*; *Falivelli et al., 2013*; *Klein and Kania, 2014*). Conditional ablation revealed that high retinal EFNA5 on nasal RGCs axons prevents temporal RGC axons from targeting the caudal SC through fiber-fiber interaction (*Suetterlin and*

*Drescher, 2014*). In our mice, such a mechanism would have generated a local duplication between *Isl2*(+) and *Isl2*(-) RGCs or an extension of the TZs in the SC which was not observed, suggesting that retinal EFNA3 is not involved in fiber-fiber interaction. However, our results do not exclude the contribution of the fiber-fiber interaction mechanism to retino-collicular map development. Together with recent results (*Suetterlin and Drescher, 2014*), our data further suggest that retinal EFNA3 and EFNA5 present distinct effects on visuotopic mapping and imply a member-specific role of retinal EFNAs in map formation. This is in contrast with retinal EPHA receptors which are considered to be functionally interchangeable (*Reber et al., 2004*; *Bevins et al., 2011*).

### Theoretical modelling further confirms mapping mechanism

Previous work modelled the stochastic nature of retino-collicular map formation based on the Koulakov model (*Tsigankov and Koulakov, 2006*; *2010*; *Owens et al., 2015*). Here, we have substantially modified and expanded this algorithm using our measured *Efna* expression data instead of theoretical values and simulated the retino- and cortico-collicular mapping process sequentially. We assumed that retinal *Efna3* expression alone cannot provide positional information, due to its homogeneous expression profile in WT RGCs. Therefore other EFNAs or other guidance molecules, either retinal or collicular, must participate in the regulation of map alignment. Retinal EFNA5 has been recently shown to participate in RGCs fiber-fiber interactions (*Suetterlin and Drescher, 2014*) but a given proportion may contribute to map alignment. We chose to retain 80% of the retinal EFNA5 level and 100% of retinal EFNA2/A3 levels in the algorithm. Moreover, contribution of endogenous collicular EFNAs to cortico-collicular map alignment seems unlikely as these mediated previous RGCs axons guidance (*Janes et al., 2005*), therefore collicular EFNA levels were not included in the V1 projection step of the model. The 3-step map alignment model replicates experimental features of both retino- and cortico-collicular maps observed in WT and Efna3 KI animals. It validates a stochastic mechanism of retinal-dependent molecular cues, involving EFNA3, providing positional information in the SC for V1 axons which then coordinate with correlated neuronal activity (*Triplett et al., 2009*) to align visuotopic maps.

These data raise important new questions as to the function of the different endogenous EFNAs in the formation of topographic maps, requiring additional development of cell-specific gene targeting approaches. From a functional standpoint, this new principle may serve as a general framework for sensory map alignment, where positional information, carried by a leading map, acts together with correlated activity, enabling precise adjustments of subsequent projection alignment.

## Materials and methods

### Generation of the Isl2-Efna3KI mice, animals and housing

The targeting construct containing a ribosomal entry site (IRES) followed by the mouse *Efna3* ORF-*SV40*polyA and the selection cassette PGK-*Neo* was inserted by homologous recombination into the 3' untranslated region of the *Islet-2* gene locus as previously performed (*Brown et al., 2000*). The mouse line was generated by the Mouse Clinic Institute, project IR3483 (Illkirch-France) in a C57/Bl6J background. Mice were hosted in a 12 hr/12 hr light-dark condition, fed ad lib. at the CNRS UMS3415 Chronobiotron (Strasbourg – France). All procedures were in accordance with national (council directive 87/848, October 1987) and European community (2010/63/EU) guidelines. Official agreement number for animal experimentation is A67-395, protocol number 01831.01 (M.R). Male and female C57/Bl6J *Efna3KI*, *Epha3KI* and *Epha4KO* mice and pups were genotyped by PCR from genomic DNA from tail biopsies as described previously (*Dottori et al., 1998*; *Reber et al., 2004*). Primers are available in Supplementary Files. Immunohistochemistry and mapping analyses were made blind to genotype.

### Projections analysis/mapping

Anterograde and retrograde DiI (1,1-dioctadecyl-3,3,3,3-tetramethylindocarbocyanine perchlorate), DiD (1,1'–dioctadecyl-3,3,3',3'- tetramethylindodicarbocyanine, 4-chlorobenzenesulfonate) and/or retrograde CTB-488 (Cholera Toxin B subunit-Alexa 488) labelling were performed as described (*Reber et al., 2004*; *Triplett et al., 2009*). Whole-mount SC were processed as described and TZs were plotted along the rostral-caudal axis on Cartesian coordinates (y axis) (*Reber et al., 2004*). For

cortico-collicular map analyses, sagittal vibratome sections were performed on P14 SC and TZs were plotted along the rostral-caudal axis on Cartesian coordinates (y axis). Retinas were dissected and imaged using Zeiss Axioscope two and Axiovision software. Retinal coordinates of the DiI injections were calculated using IntactEye algorithm (*Hjorth et al., 2015*), confirmed using the projection method (*Reber et al., 2004*) and plotted on Cartesian coordinates (x axis). V1 cortices were photographed as whole-mount and focal injections plotted along the V1 lateral-medial axis (x axis) (*Triplett et al., 2009*). Retino-collicular and cortico-collicular maps were generated using non-parametric smoothing technique, termed LOESS smoothing (*Efron and Tibshirani, 1991*), to estimate the profile of the one-dimensional mapping either from retina to SC, or from V1 to SC. To estimate the variability in a mapping containing N data points, we repeated the procedure N times with N-1 datapoints, each time dropping a different datapoint. This is termed a 'leave-one-out' method and was used in the R Project for Statistical Computing (RRID:SCR_001905). The script is available in *Source code 1* file. Retino-geniculate and binocular tracings were performed as described (*Pfeiffenberger et al., 2006*) on Efna3KI/KI (n = 4) and WT (n = 3) P7 animals. Analyses of the cortico-collicular projections were performed using the thresholding function in ImageJ (RRID:SCR_003070).

## Immunohistochemistry

Retinas, ONs, SCs and V1 cortices were dissected after animal perfusion with PFA4% in PBS1X, post-fixed O/N in PFA4%/PBS1X, cryoprotected in PFA4%/sucrose 30%/PBS1X for several hours at 4°C. Retinas and ONs were cryostat-sectioned (14–18 µm), SCs and V1 cortices were vibratome-sectioned (40 µm) and processed for immunohistochemistry. Experiments were performed at least three times. Briefly, sections were incubated in blocking solution (PBS 1X/BSA 1%/normal goat serum 10%) for 1 hr at RT then incubated with primary antibody O/N at 4°C in PBS 1X/BSA 1%/normal goat serum 1%. The following day, sections were washed (3 × 5' in PBS 1X at RT) and incubated for 1 hr at RT with secondary Alexa-labelled antibodies in PBS1X/BSA 1%/normal goat serum 1%. After three washes at RT (3 × 5' PBS 1X), slides were mounted in Aqua-Polymount (Polysciences Europe GmbH, Eppelheim, Germany) and visualized under a confocal microscope (Leica SP5 II, Leica Microsystems, Wetzlar, Germany). Sections were imaged using Leica LASAF software. Antibodies: anti-Isl2 (1/400, ref. LS-C165303, LifeSpan Biosciences Inc., RRID:AB_2126601), anti-Efna3 (1/300, ref. LS-C6547, LifeSpan Biosciences Inc., RRID:AB_797145), anti-EphA3 (1/100, ref. LS-C150188, LifeSpan Biosciences Inc., RRID:AB_11187174), anti-Efna3 (ref. 36–7500, 1/200, Invitrogen, Invitrogen Co. RRID:AB_2533278), anti-synaptophysin (1/200, ref. S5768, Sigma-Aldrich), anti-Tau (1/200, MAB3420, Merck Millipore), anti-NF200 (1/200, ref. N5389, Sigma-Aldrich), anti-rabbit Alexa 488 (RRID:AB_10373124), anti-goat Alexa 594 (RRID:AB_10562717), anti-mouse Alexa 555 (RRID:AB_10561552) (1/500, Invitrogen, Invitrogen Co.).

## Quantitative RT-PCR

V1 cortices, superficial layers of the SC and retinas were freshly dissected. Retinas were cut in three equal pieces along the NT axis (Nasal, Central, Temporal RGCs) and RGCs acutely isolated (*Steinmetz et al., 2006*; *Claudepierre et al., 2008*). Total RNA was extracted and quantified as previously described (*Mathis et al., 2015*). Briefly, relative quantification was performed using the comparative Delta Ct method. Triplicates were run for each sample and concentration for the target gene and for two housekeeping genes (hypoxanthine-guanine phosphoribosyl transferase - *Hprt* and glyceraldehyde 3-phosphate dehydrogenase – *Gapdh*) were computed. Primers are listed in Supplementary File.

## Retinal ganglion cell isolation

P3/P4 retinas were freshly dissected and RGCs were isolated and purified (>99%). For details see *Steinmetz et al. (2006)*; *Claudepierre et al., 2008*). Briefly, cells were cultured in Neurobasal medium (Gibco/Invitrogen) supplemented with (all from Sigma, except where indicated) pyruvate (1 mM), glutamine (2 mM; Gibco/Invitrogen), *N*-acetyl-l-cysteine (60 µg ml$^{-1}$), putrescine (16 µg ml$^{-1}$), selenite (40 ng ml$^{-1}$), bovine serum albumin (100 µg ml$^{-1}$; fraction V, crystalline grade), streptomycin (100 µg ml$^{-1}$), penicillin (100 U ml$^{-1}$), triiodothyronine (40 ng ml$^{-1}$), holotransferrin (100 µg ml$^{-1}$), insulin (5 µg ml$^{-1}$) and progesterone (62 ng ml$^{-1}$), B27 (1:50, Gibco/Invitrogen), brain-derived

neurotrophic factor (BDNF; 25 ng ml$^{-1}$; PeproTech, London, UK), ciliary neurotrophic factor (CNTF; 10 ng ml$^{-1}$; PeproTech) and forskolin (10 μm; Sigma). After isolation, RGCs were either treated for RNA extraction or fixed with PFA4% 15′ at RT and processed for immunohistochemistry. Stainings were performed and cells were visualized as described above.

### Semi-quantitative in situ hybridization and gradient fitting of retinal Efnas expression

Analysis of the expression of the *Efnas* was performed as previously described (*Reber et al., 2004*) on nasal-temporal 20 μm thick sections of P1/P2 WT retinas. Probes used were: mouse *Efna2* (NM007909.3, 879 bp, pos. 387–1266), mouse *Efna3* (NM010108, 791 bp, pos. 208–999) and *Efna5* (NM207654, 696 bp, pos. 189–885). Experimental values (mean +SD, *Efna2/a3/a5*, for each *Efnas*, n = 18 sections total, 3 sections/retina, from 6 retinas −2 left, 2 right- from 3 animals) were plotted along the nasal-temporal axis and fitted using MATLAB (RRID:SCR_001622).

### In silico replication of the duplication of the cortico-collicular map

The Koulakov model in MATLAB (RRID:SCR_001622) (*Tsigankov and Koulakov, 2006*; *2010*) was used to simulate the formation of both the retino- and cortico-collicular maps in the presence of an oscillatory *Efna* gradients in the target structure. Each brain structure (retina, SC, V1) is modelled as a 1-d array of 100 neurons (N) in each network. Two maps are generated: first, the map from retina to SC; second, the map from V1 to SC. Each map is modelled sequentially in the same way. This model consists in the minimization of affinity potential (E) which is computed as follows:

$$E = E_{act} + E_{chem}$$

At each step, this potential is minimized by switching two randomly chosen axons probabilistically according to the degree such a switch reduces the energy in the system by Delta E ($\Delta$E). The probability of switching, p, is given by: p=1/(1 + exp(4$\Delta$E)).

$E_{chem}$ is expressed as follows:

$$E_{chem} = \sum\nolimits_{i \in TZS} \alpha [R_A(i) - R_A(j)][L_A(i') - L_A(j')]$$

where $\alpha$ is the strength ($\alpha$ = 200), $R_A$(i) and $R_A$(j) the receptor concentration in the retina and V1 at location (i) and (j) and $L_A$(i′) and $L_A$(j′) the ligand concentration at the corresponding position (i′) and (j′) in the SC. The contribution of activity-dependent process is modelled as:

$$E_{act} = -\gamma/2 \sum\nolimits_{ij \in synapses} C_{ij} U(r)$$

where $\gamma$ = 1 is the strength parameter, $C_{ij}$ is the cross-correlation of neuronal activity between two RGCs (or V1) neurons during spontaneous activity located in (i) and (j), and U simulates the overlap between two SC cells. Here, we use $C_{ij}$ = exp(−r/R), where r is the retinal distance between RGC (i) and (j), R = 0.11 × N, and U(r′) = exp(−r′$^2$/2d$^2$), where r′ is the distance between two SC points (i′, j′), d = 3 and N = 1 to 100 neurons.

Receptor and ligand gradients were modelled as follows:

Retinal *Epha* gradients (*Reber et al., 2004*):

$$R_A(x)^{retina} = 0.26^* exp(0.023x) + 1.05$$

Cortical V1 *EphA* gradients (*Tsigankov and Koulakov, 2006*; *2010*):

$$R_A(x)^{V1} = exp(-x/N) - exp((x - 2N/(N)) + 1)$$

Collicular *Efna* gradients (*Tsigankov and Koulakov, 2006*; *2010*):

$$L_A(x)^{SC} = exp((x - N)/N) - exp((-x - N)/N)$$

$L_A$(x)$^{retina}$ is the *Efna* gradient which was modelled by an exponential fitting of the in situ hybridization data. These retinal *Efna* gradients (80% of *Efna5* expression and 100% of *Efna2* and *Efna3*) are transposed to the SC for the simulation of the cortico-collicular map.

Retinal *Efna* gradients equation along the nasal (0) – temporal (100) axis (see Results):

$$L_A(x)^{\text{retina WT}} = L_{A5}(x)^{\text{retina}} + L_{A2}(x)^{\text{retina}} + L_{A3}^{\text{retina}}$$

$$L_A(x)^{\text{retina WT}} = (1/0.56) * \exp(-0.014x) + (1/0.54) * \exp(-0.008x) + 0.44$$

(*Figure 7*) where x = 1 to N and N = 100 representing the neurons along each axes (L-M in V1, N-T in retina). When transposed along the rostral-caudal axis in the SC (retina → SC), retinal *Efna* gradients are flipped along the x axis and become:

$$L_A(x)^{\text{retina} \to \text{SC}} = (1/0.56)^* \exp(-0.014(100-x)) + (1/0.54)^* \exp(-0.008(100-x)) + 0.44$$

$$= 0.56^* \exp(0.014x) + 0.54^* \exp(0.008x) + 0.44$$

The oscillatory gradient was generated by randomly attributing to 50% of collicular TZs an over-expression of *Efna3* ($\Delta L_{A3}$) with $\Delta L_{A3}$ = 0.44 for homozygotes and $\Delta L_{A3}$ = 0.22 for heterozygotes. Iterations were ran for $10^7$ epochs. 3-step map alignment model scripts can be found in Source Code two file.

## Quantitative analysis of theoretical maps

To determine the amount of duplication that could be found in heterozygotes and homozygotes mutants a linear regression was calculated using implemented functions in Matlab. The residuals were then used to calculate the percentage of duplication. Duplications were considered when values were outside EP = 9.18% for Efna3KI/+ and EP = 15.18% for Efna3KI/KI, which corresponds to the averaged experimental distances measured between duplicated termination zones ($\Delta S_{exp}$) to which was added the wild-type map variability (average of residuals, $\sigma_{WT}$ = 2.18 %, n = 20 runs). Twenty runs were performed and averaged to find the proportion of duplicated termination zones for Efna3 KI.

## Data/codes availability

Software package IntactEye can be found at: http://github.com/hjorthmedh/IntactEye. 'Leave-one-out' script in R and 3-step Map Alignment Model provided as a source code files (*Source code 1* and *2*). Model also available at *Reber, 2017* (with a copy archived at https://github.com/elifescien-ces-publications/3-step-Map-Aligment-Model).

## Acknowledgements

We are grateful to Dr. Feldheim for providing biological material from Efna3KO mice. We thank Dr. Sophie Reibel-Foisset, Laurence Huck and Noémie Charléry-Adele (UMS 3415 CNRS – Chronobio-tron, Strasbourg, France) for mouse colony management, Dr. Nick Bevins and Dr. Alexandre Charlet for technical help, Dr. Melinda Owens for providing the Koulakov/Triplett scripts in MATLAB and Dr. Alexandra Rebsam for providing a batch of DiD. ES, MP, AB and MR performed the experiments. SE generated the Leave-One-Out algorithm. GL and MR analyzed the Efnas ISH quantitation. ES, S.E and MR generated the theoretical modelling. ES, SE FWP and MR analyzed the results. MR wrote the manuscript with the active participation of all the authors. Material requests and correspondence should be addressed to MR email: michael.reber@inserm.fr. This work was supported by CNRS and University of Strasbourg – Institute for Advanced Study (MR).

# Additional information

## Funding

| Funder | Grant reference number | Author |
| --- | --- | --- |
| Centre National de la Recherche Scientifique | recurent yearly funding 2013-2016 | Frank W Pfrieger Michael Reber |
| Université de Strasbourg | recurent yearly funding 2013-2016 | Frank W Pfrieger Michael Reber |

| Université de Strasbourg | Institute for Advanced Study - Escellence Program Fellowship 2012-2015 | Stephen J Eglen Michael Reber |

The funders had no role in study design, data collection and interpretation, or the decision to submit the work for publication.

## Author contributions
ES, Data curation, Software, Formal analysis, Validation, Methodology; SJE, Conceptualization, Software, Validation, Methodology, Writing—review and editing; AB, MP, Data curation, Methodology; FWP, Conceptualization, Resources, Methodology, Writing—review and editing; GL, Resources, Methodology, Writing—review and editing; MR, Conceptualization, Resources, Data curation, Software, Formal analysis, Supervision, Funding acquisition, Validation, Investigation, Visualization, Methodology, Writing—original draft, Project administration, Writing—review and editing

## Author ORCIDs
Elise Savier, http://orcid.org/0000-0001-7512-1630
Stephen J Eglen, http://orcid.org/0000-0001-8607-8025
Michael Reber, http://orcid.org/0000-0001-8842-7276

## Ethics
Animal experimentation: All animal procedures were in accordance with national (council directive 87/848, October 1987) and European community (2010/63/EU) guidelines. Official agreement number for animal experimentation is A67-395, protocol number 01831.01 (M.R).

## Additional files

### Supplementary files
• Source code 1. Leave-one-out 'Leave-one-out' script in R.

• Source code 2. 3-step map alignment model 3-step map alignment script in MATLAB

• Supplementary file 1. Sequences of the primers used for genotyping and expression analyses.

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
