## [Decision Letter]

[Editors’ note: this article was originally rejected after discussions between the reviewers, but the authors were invited to resubmit after an appeal against the decision.]

Thank you for submitting your work entitled "A Molecular Mechanism for the Topographic Alignment of Convergent Neural Maps" for consideration by *eLife*. Your article has been reviewed by three peer reviewers, and the evaluation has been overseen by a Reviewing Editor and by Andrew King as Senior Editor. The reviewers have opted to remain anonymous.

Our decision has been reached after consultation with the reviewers. Based on these discussions among the reviewers and the individual reviews below, we regret to inform you that your work will not be considered further for publication in *eLife* at this time.

All three reviewers considered your study important and felt it would be a valuable addition to the field. They described your finding that expression of ectopic ephrin-A3 in *Isl2(+)* RGCs creates a discontinuous gradient of ephrin-A3 on RGC axons, and that this provides repellent cues to guide V1 axons to topographically correct locations in the SC, as novel and potentially exciting.

The reviewers nonetheless had several suggestions for improvements that would strengthen the study, some of which will likely involve additional and somewhat lengthy experiments:

1) Demonstration of the discontinuous gradient of ephrin-A3 protein in the SC. This may be understandably difficult to do if the antibodies are not good.

2) In line with the discontinuity demonstration: a two-color V1 injection that would enable visualization of interdigitated red and green termination zones (TZs) in the SC. This would fortify your conclusions.

3) The figure/micrograph quality is sub-par. One suggestion for getting a better read on duplication of the cortical maps would be to section the SC and to show the results at higher magnification than used in the current version, rather than showing surface images.

4) in vitro assays as an alternative for demonstrating ki-ephrin-A3 on axons by checking for altered sensitivity of ephrin-A3 KI/KI axons to EphA3-Fc, using a collapse or similar assay.

5) Revision of the assumptions and other aspects of the modeling, as indicated in reviewer 3's comments.

*Reviewer #1:*

Savier et al. describe the changes in retinocollicular and V1-collicular maps when ectopic ephrin-A3 is expressed in *Isl2(+)* RGCs, creating a discontinuous gradient of ephrin-A3 on RGC axons. This is a similar approach to that used by Brown et al. 2000 and Triplett et al. 2009 that showed that to ectopically expressed EphA3 in *Isl2(+)* RGCs leads to map duplications in the retinocollicular and V1-SC projections. Unlike when EphA3 is ectopically expressed in the retina, the authors find that ectopic expression of ephrin-A3 does not affect the topography of the retinocollicular projection. Interestingly, retinal expressed ephrin-A3 leads to changes in the V1-SC topographic map. These results lead the authors to a computational model whereby RGC axons in the SC provide repellent cues to guide V1-SC axons to topographically correct locations.

This work is very interesting and timely and the authors propose a compelling model to explain the data in the context of topographic map alignment.

1) I am confused why ectopic expression of ephrin-A3 in *Isl2(+)* RGCs does not lower the effective concentration of EphAs in these cells. The experiments in Figure 6 show that extra EphA3 can mask the ectopic ephrin-A3. So why doesn't the ectopic ephrin-A3 mask the available EphA, especially in the nasal half of the retina where EphA levels are graded and low?

The authors address this in the Discussion "However, we did not observe any retino collicular mapping defects regardless of the mutant genotypes, suggesting that ephrin-A3 ectopic expression does not inactivate endogenous co-expressed EphA4/A5/A6 receptors in RGCs, although we showed specific inactivation of EphA3, suggesting distinctive interactions in cis between ephrin-A/EphA pairs as previously observed (Yin et al., 2004; Falivelli et al., 2013; Klein and Kania, 2014)." But this explanation is somewhat unsatisfying based on the current knowledge of ephrin-A binding affinities for EphA receptors.

It would be nice if the authors can address the specificity of masking more directly, perhaps by showing ectopic ephrin-A3 does not change the sensitivity of RGC axons to ephrin-A5 in trans, although I understand that this may be beyond the scope of the paper.

2) It would be nice to directly visualize the discontinuity of ephrin-A3 expression on RGC axons in the SC using immunocytochemistry.

3) The duplication of the V1-SC projection is not very pronounced. It seems possible that the termination zones are just a bit blurry. Two color DiI, DiO injections in V1 would unambiguously demonstrate that the TZs are duplicated (see Triplett et al. 2009 Figure 6).

4) The SC s of the anterograde DiI retinal injections should be sectioned as done for the V1-SC projection. It is possible that ectopic termination zones in deeper SC lamina may not be revealed in a whole mount view.

*Reviewer #2:*

The manuscript by Savier et al. describes a novel ephrinA3-based mechanism that guides the alignment of convergent topographic maps in the mouse superior colliculus. These findings add to a rich literature of how gradients of Eph Receptors and their ligands in the retina and brain lead to topographic maps. An interesting twist on the studies presented here is that retinal ganglion cells generate ephrin A3, and its presence on retinal terminals (or arbors) in the SC contribute to the mapping of cortical projections from V1 that converge on target cells in the SC. Several other sets of studies have now shown how one class of axons in a given visual target, dictate how other classes or types of axons innervate that same target (Triplett et al. 2009; Seabrook et al. 2013; Brooks et al. 2013; Grant et al. 2016; Shanks et al. 2016). That being said, little is known about what molecular cues on an axon may dictate this and for this reason this paper will be of interest to the field.

There are several strengths to this manuscript, including the generation of a novel mouse mutant in which ephrinA3 is over expressed in a subset of RGCs. This approach was previously used to misexpress EphA3, which led to the discovery that EphA3 overexpression leads to a duplication of retinocollicular maps. Interestingly, perturbing ephrinA3 expression in RGCs had little affect on retinocollicular or retinogeniculate circuits. Another strength of the manuscript is that the authors use a clever trick to inactivate ephrins and EphA3 by co-expressing a single allele each under the Islet2 promoter and they develop a novel computation model to support their experimental data.

There are, however, some weaknesses in the manuscript that require additional data and analyses. Without these changes this reviewer does not think this work is suitable for publication in *eLife*.

These major concerns include:

Is there an alteration of ephrinA3 distribution in the SC of the *Isl2(-)* ephrin-A3KI mice? EphrinA3 IHC was performed in this manuscript, but only in retina. For the model proposed to be valid (really the crux of the manuscript) one would expect to see fluctuations in ephrin A3 distribution in SC in mutants (associated with retinal arbors). This is an essential piece of data that is missing. And it should be done on WT, single copy mutants, and double copy mutants.

The duplications of cortical maps are quite subtle in these mutants, compared to those of previous studies. In fact, they are so subtle they are difficult to discern in some cases in the manuscript – the images in Figure 4 both look like double maps compared to WT controls. How were double maps quantified and identified? It seems critical to show the injection sites in V1 for each case (like is done for retinocollicular maps) to ensure that double maps are not the result of wider injection sites. Likewise, an important part of the manuscript was that retinal maps are unaffected. But the analysis of those maps seems less detailed than the cortical maps. Perhaps subtle changes in retinal maps would be observed if sections of SC were analyzed at high magnification, rather than surface images. Again, consistency in analysis methods for both sets of maps and data is important for the conclusions of the manuscript.

While the inactivation of ephrinA3 overexpression by co-expression of EphA3 was viewed as a strength of the manuscript, some of the data demonstrating the validity of this approach was dumped into the supplemental material. It seems too important to be "supplemental" as it is a critical component of this manuscript. Likewise more data needs to be presented to convince this reviewer that this result is not the result of the lack of penetrance in single copy of EphA3 mutant, since it has been previously reported that duplicated maps are observed in only some single copy mutants.

In addition to major concerns described above, there are also concerns in the presentation of the manuscript:

1) In some cases, images appear to be poorly selected to make the authors point. For example see Figure 2 which are described as being similar but are not of equal quality. Other examples include 3H-K, Figure 4, and Figure 6.

2) In several cases proper labeling is missing or inappropriate on the figures. In Figure 1 labels are missing. In Figure 3 what are the arrows? (And the figures in Figure 3 are insufficient to claim normal eye-specific segregation in this mutant). It’s unclear what Figure 5 is amplifying? (And if its ephrinA3 why isn't it in WT retina?). More care should be paid to all figures and legends.

*Reviewer #3:*

The formation of topographic axonal maps is a key element of brain wiring. Visuotopic mapping is mainly instructed by the ephrin/Eph system, enabling forward, reverse, axon/target, axon/axon and cis signals. The actual contribution of each of these signals and their integration, has remained highly controversial. The authors specifically address the role of axonally located ephrin-As by a knock-in (KI) of ephrin-A3 into a random half-population of retinal ganglion cells (RGCs). Therefore, their paper is of major importance to the field. Their approach is valid, the experiments are thoroughly performed, carefully evaluated and lucidly presented in the paper. I have, however, two (somewhat) major issues that should be clarified before publication.

Based on our current understanding, a duplicated retinocollicular map in the KI/KI animals would be expected. Ki axons should grow to the posterior superior colliculus (SC) due to axon/target reverse signaling (Rashid T., et al. 2005). Wildtype axons should be anteriorly displaced due to axon/axon forward signaling (Suetterin P., et al., 2015). Intriguingly, the retinocollicular map is unaffected by the ki. In contrast, however, the V1-corticocollicular map is split up, notably indicating a role of ephrin-A3 on retinal axons in the chemotopic guidance of V1 cortical axons. The same projection has previously been claimed to be aligned by activity-dependent mechanisms (Triplett J., et al., 2009). Although very exciting, the new results are to some degree in conflict with previous observations, necessitating thorough control. Thus, I'm still not fully convinced, that an amount of KI ephrin-A3 arrives in the SC that could be functionally relevant to retinocollicular mapping. Maybe the pioneering retinocollicular mapping is more robust to changes than the following corticocollicular map. Immunostainings of isolated kiRGCs display labeling on the neurites (Figure 1'), but this is after 6 DIV and it remains unclear whether the shown neurites are axons or dendrites. The authors show that concomitant KI of EphA3 rescues the mapping effects of both knock-ins, but the mutual neutralization of ephrin-A3 and EphA3 might already occur in the retina. Therefore, the authors should provide some additional evidence for the axonal localization of the KI ephrin-a3 in the SC (e.g., Western blots, immunostainings on the SC or in-vitro assays).

My second major issue relates to the modeling. The presented model principally aids the understanding of the corticocollicular effects. In its present state, however, it has some deficiencies. First, either the key equation of the model, given in subsection “In silico replication of the duplication of the cortico-collicular map”, referring to two gradients on both, the target and the field of origin is wrong, or not all gradients used are specified. Second, I can't see how the fitting equations given for the ephrin gradients relate to the actual measurements. And third, for modeling corticocollicular mapping, the authors arbitrarily reduce the ephrin-A5 gradient to 80%. This is neither well motivated, nor has it been evaluated, how this affects the outcome. When the mentioned issues have been clarified, I will cordially support the publication of this nice piece of work in *eLife*.

[Editors’ note: what now follows is the decision letter after the authors submitted for further consideration.]

Thank you for submitting your article "A Molecular Mechanism for the Topographic Alignment of Convergent Neural Maps" for consideration by *eLife*. Your article has been reviewed by three peer reviewers, and the evaluation has been overseen by a Reviewing Editor and Andrew King as the Senior Editor. The reviewers have opted to remain anonymous.

The reviewers have discussed the reviews with one another and the Reviewing Editor has drafted this decision to help you prepare a revised submission.

Summary:

All three reviewers agree that you and your co-authors have nicely addressed the criticisms and revised the figures, including demonstration of duplicated maps, appropriately and made other requested revisions. The reviewers reiterated that they consider this an important study for the field.

In their appraisal of the initial manuscript, the reviewers called for a demonstration of RGC-expressed ephrin A3 to support the main tenet of the study – that RGC axon-derived ephrin A3 plays a role in shaping corticocollicular (CC) maps. For the revised manuscript, the reviewers appreciated the new data showing localization with the commercial antibody from LSBio, with good staining of RGCs and expression on axon-like profiles in the optic nerve in the proximal and middle (erroneously termed "medial") sectors, albeit not in the distal sector. We acknowledge that immunohistochemistry was the most expedient action to take for the demonstration of ephrin A3, even though most in the field have failed to reveal Ephs and ephrins by this method. Nonetheless, we wish to see additional revisions for the ephrin A3 expression as listed below, first and foremost, to demonstrate antibody specificity.

Essential revisions:

1) Confirm antibody specificity by staining WT vs. ephrin-A3 KO retinal and SC tissue, if possible testing tissue from the same litter (or source/background). This would validate the results, especially for the expression in the retina and optic nerve.

2) Include results of the expression as shown in the rebuttal in the manuscript itself, in Supplemental Data.

3) More thoroughly address the "dissonance" in ephrin-A3 expression along RGC axons in the Discussion.

4) Ephrin-A3 is present on axons, but below the detection limit; thus, there cannot be 100% transfer of the retinal ephrin-A3 into the axons. You should include this finding in your model.

---

## [Author Response]

Editors’ note: the author responses to the first round of peer review follow.]

*[…] Reviewer #1: […] 1) I am confused why ectopic expression of ephrin-A3 in Isl2(+) RGCs does not lower the effective concentration of EphAs in these cells. The experiments in Figure 6 show that extra EphA3 can mask the ectopic ephrin-A3. So why doesn't the ectopic ephrin-A3 mask the available EphA, especially in the nasal half of the retina where EphA levels are graded and low? The authors address this in the Discussion "However, we did not observe any retinocollicular mapping defects regardless of the mutant genotypes, suggesting that ephrin-A3 ectopic expression does not inactivate endogenous co-expressed EphA4/A5/A6 receptors in RGCs, although we showed specific inactivation of EphA3, suggesting distinctive interactions in cis between ephrin-A/EphA pairs as previously observed (Yin et al., 2004; Falivelli et al., 2013; Klein and Kania, 2014)." But this explanation is somewhat unsatisfying based on the current knowledge of ephrin-A binding affinities for EphA receptors. It would be nice if the authors can address the specificity of masking more directly, perhaps by showing ectopic ephrin-A3 does not change the sensitivity of RGC axons to ephrin-A5 in trans, although I understand that this may be beyond the scope of the paper.*

Binding affinities for EphAs and ephrin-As have indeed been documented (Brambilla et al., 1995; Gale et al., 1996), but assess binding of EphA receptors and ephrin-A ligands in trans (i.e., on different cells or applied exogenously). These values do not apply here as our hypothesis suggests an interaction in CIS (on the same cell), for which binding affinities data are unavailable. Our results suggest a cis-inactivation of EphA3 and ephrin-A3 when expressed on the same RGCs, but not with endogenous retinal EphA4/A5/A6 receptors. CIS-inactivation of the EphA4/A5/A6 endogenous receptors by ectopic ephrin-A3 would affect their binding to collicular ephrin-As. In the ephrin-A3KI mutants, this would generate 2 populations of RGCs: one, the *Isl2(-)*, presenting WT EphAs binding capacity in TRANS and the other, the *Isl2(+)*, with decreased EphAs binding capacity in TRANS. In such a scenario and according to the Relative Signaling mechanism (Brown et al., 2000; Reber et al., 2004; Bevins et al., 2011) a duplicated (full or partial) retino-collicular map should be obtained, which was not observed (refer to Figure 3 of our revised version). To further support our interpretation, we characterized the retino-collicular maps in compound mutant ephrin-A3KI/KI, EphA4KO. In these mutants, decreasing the relative EphA signaling strength, by diminishing or abolishing homogeneous EphA4 retinal expression, would reveal any EphA4/A5/A6 signaling defect (a consequence of a cis-inactivation) upon collicular ephrin-As binding in TRANS. As a consequence, duplicated retino-collicular maps should be observed in the compound mutants, particularly in the caudal pole of the SC where nasal RGCs, expressing low levels of EphAs, project. However, as shown in Figure 10, our data revealed normal retino-collicular maps in ephrin-A3KI/KI, *EphA4+/-* and ephrin-A3KI/KI, *EphA4-/-*, indicating that ectopic expression of ephrin-A3 in *Isl2(+)* RGCs does not inactivate endogenous retinal EphAs receptors.

Author response image 1.Retino-collicular projections analysis at P8 in *ephrin-A3KI/KI∷EphA4+/-* (**A**) and *ephrin-A3KI/KI∷EphA4-/-* (**B**) double mutants showing single retino-collicular maps (left panels) and examples of anterograde tracings (right panels: whole-mount confocal SCs images showing the single TZs – white spot).Scale bars: 200 µm. R, rostral; C, caudal.**DOI:**
http://dx.doi.org/10.7554/eLife.20470.023

*2) It would be nice to directly visualize the discontinuity of ephrin-A3 expression on RGC axons in the SC using immunocytochemistry.*

The discontinuity (or alternating pattern) of ephrin-A3 expression on RGCs in the SC is generated by the homogeneous distribution of the *Isl2(+)* and *Isl2(-)* RGC TZs forming the retino- collicular map. To visualize the discontinuity of ephrin-A3 expression on RGCs axons /TZs in the SC of ephrin-A3KI/KI animals, we performed an immunohistochemical analysis, using the ephrin-A3 antibodies that clearly revealed ectopic ephrin-A3 expression in the retina (LSBio LS- C6547, Figure 1). Our data show a strong ephrin-A3 staining level in collicular cells, similar between WT and ephrin-A3KI/KI littermates at P3 and P8 (in accordance with qPCR data shown Figure 5 in our revised version). However, no discontinuity in ephrin-A3 staining in the SC can be observed in ephrin-A3KI/KI. This is likely a consequence of the exceptionally high level of *endogenous* ephrin-A3 staining in collicular cells, making any additional ephrin-A3 variation on RGCs axons difficult to discern immunohistochemically.

Author response image 2.Immunohistochemical staining on P3 and P8 ephrin-A3KI/KI and WT SC parasagittal sections revealing ephrin-A3 expression in collicular cells (controls are without primary anti-ephrin-A3 antibody).Scale bars: 200 µm (A – D), 30 µm (A’ – D’, A’’, B’’); R, rostral; C, caudal.**DOI:**
http://dx.doi.org/10.7554/eLife.20470.024

We further analyzed ephrin-A3 expression using immunohistochemistry (amplified by streptavidin/biotin) on RGCs axons in parasagittal sections of the SC after DiI labelling of RGCs at P3 and P8. As shown in Figure 12, DiI labelled RGC axons can be observed in the SC, as well as strong collicular ephrin-A3 staining. However, no ephrin-A3 staining on these very thin DiI- labelled RGC axons can be detected.

Author response image 3.Immunostaining revealing ephrin-A3 expression (green) in P3 and P8 ephrin- A3KI/KI and WT SC parasagittal sections after retinal anterograde DiI injection (red fibers).Scale bars: 30 µm (A, D), 10 µm (B – F).**DOI:**
http://dx.doi.org/10.7554/eLife.20470.025

We next performed ephrin-A3 immunohistochemistry on P3 ephrin-A3KI/KI and WT optic nerve longitudinal sections (proximal, medial, and distal to the optic cup) using signal amplification (streptavidin/biotin). As shown in Figure 13, specific ephrin-A3 labeling could be detected readily on fibers within the proximal part of the optic nerve in WT and ephrin-A3KI/KI mutants, and the medial part of the optic nerve in ephrin-A3KI/KI, but was nearly impossible to discern in the distal part, suggesting that ephrin-A3 is indeed present on RGCs axons (as shown in vivo on retinal sections Figure 1 and in vitro Figure 1’ of the revised version), but that expression becomes dispersed as these axons advance into the SC. The absence of strong staining in the distal part in ephrin-A3KI/KI and WT suggests that ephrin-A3 ligands are spread along the axons and reach a low concentration, below the detection limit of immunohistochemistry. This would explain why ephrin-A3 cannot be detected on RGC axons within the SC (see response above §2).

Author response image 4.P3 ephrin-A3KI/KI (A-C’, G, G’) and WT (D-F’) optic nerve sections, at the proximal, medial and distal level from the optic disc, immunostained for ephrin-A3 (in green), DAPI (in blue) after retinal DiI injections (in red).A control experiment (without ephrin-A3 antibody) is shown on ephrin-A3KI/KI proximal optic nerve section. Scale bars: 20 µm (A-G’), 10 µm (high magnification panels in A’, B’, D’, E’). Arrows in insets show ephrin-A3 positive fibers.**DOI:**
http://dx.doi.org/10.7554/eLife.20470.026

*3) The duplication of the V1-SC projection is not very pronounced. It seems possible that the termination zones are just a bit blurry. Two color DiI, DiO injections in V1 would unambiguously demonstrate that the TZs are duplicated (see Triplett et al. 2009 Figure 6).*

As requested by reviewer 1, two color injections in V1 (red: DiI, cyan: DiD) were performed. As shown in Figure 4—figure supplement 2 (subsection “Cortico-collicular maps are duplicated in Isl2-Efna3KI mutants”, first paragraph), duplicated cortico-collicular TZs can be revealed at P15 in ephrin-A3KI/KI animals, whereas WT littermates only show single projections. Corresponding V1 injections are shown as whole-mount (white spot: DiI; dark spot: DiD).

*4) The SC s of the anterograde DiI retinal injections should be sectioned as done for the V1-SC projection. It is possible that ectopic termination zones in deeper SC lamina may not be revealed in a whole mount view.*

Analyses of the retino-collicular TZs by sagittal sectioning were originally performed at the time of the retino-collicular maps characterization. Figure 3—figure supplement 2 (subsection “Knock-in mice for Efna3 ectopic expression in Isl2(+) RGCs show normal retino-collicular and retino-geniculate projections”, last paragraph) shows examples of collicular sagittal sections after anterograde retinal labelling in WT and ephrin-A3KI/KI SCs at P8. Projections in both genotypes are similar in terms of layering, size and density of the TZs.

*Reviewer #2: […] Is there an alteration of ephrinA3 distribution in the SC of the Isl2(-) ephrin-A3KI mice? EphrinA3 IHC was performed in this manuscript, but only in retina. For the model proposed to be valid (really the crux of the manuscript) one would expect to see fluctuations in ephrin A3 distribution in SC in mutants (associated with retinal arbors). This is an essential piece of data that is missing. And it should be done on WT, single copy mutants, and double copy mutants.*

This same point was made by reviewer 1. Please see our comments and data as above. Immunohistochemistry on WT and ephrin-A3KI mutant littermates was performed (see response above, reviewer 1 §2). They revealed very strong endogenous ephrin-A3 staining in collicular cells at P3 and P8 in both eprin-A3KI/KI and WT animals. However, no alternating pattern above this staining pattern can be observed in ephrin-A3KI/KI when compared to WT littermates. As mentioned above (see response reviewer 1 §2 and Author response Figure 2–Figure 4), we believe this represents a signal-to-noise issue that reflects (a) the strong ephrin-A3 staining in collicular cells masking any ephrin-A3 variation on RGCs axons, and (b) the relatively low level of knock- in ephrin-A3 present on the very thin retinal axons, which is below the detection level afforded by immunohistochemistry. As outlined above, there is knock-in ephrin-A3 expression in the optic nerve that is readily detectable by immunohistochemical methods.

*The duplications of cortical maps are quite subtle in these mutants, compared to those of previous studies. In fact, they are so subtle they are difficult to discern in some cases in the manuscript – the images in Figure 4 both look like double maps compared to WT controls. How were double maps quantified and identified? It seems critical to show the injection sites in V1 for each case (like is done for retinocollicular maps) to ensure that double maps are not the result of wider injection sites.*

To convince the reviewer that V1-SC projections are indeed duplicated in ephrin-A3KI mutants, we provide examples of pictures from sagittal sections of the SC after V1 DiI labelling in both ephrin-A3KI/KI and ephrin-A3KI/+ mutants as well as WT littermates. Thresholding using ImageJ was performed with the same parameters for all 3 genotypes and revealed single and duplicated cortico-collicular TZs in ephrin-A3 mutants and only single TZs in WT littermates.

These figures were added to our revised version, Figure 4—figure supplement 1 and subsection “Cortico-collicular maps are duplicated in *Isl2(-)* Efna3KI mutants”, first paragraph. We also added whole-mount pictures of the V1 injections corresponding to the duplicated and single cortico-collicular TZs shown in our revised version of Figure 4 and Figure 4 legend.

*Likewise, an important part of the manuscript was that retinal maps are unaffected. But the analysis of those maps seems less detailed than the cortical maps. Perhaps subtle changes in retinal maps would be observed if sections of SC were analyzed at high magnification, rather than surface images. Again, consistency in analysis methods for both sets of maps and data is important for the conclusions of the manuscript.*

Detailed analysis of retino-collicular TZs in ephrin-A3KI/KI mutants have been performed using sagittal section of the SC and confocal microscopy (see response to reviewer 1 §4 and Figure 3—figure supplement 2 and subsection “subsection “Knock-in mice for Efna3 ectopic expression in Isl2(+) RGCs show normal retino-collicular and retino-geniculate projections”, last paragraph”) and revealed no major differences (in size and layering of the TZs) compared to WT littermates.

*While the inactivation of ephrinA3 overexpression by co-expression of EphA3 was viewed as a strengt, of the manuscript, some of the data demonstrating the validity of this approach was dumped into the supplemental material. It seems too important to be "supplemental" as it is a critical component of this manuscript. Likewise more data needs to be presented to convince this reviewer that this result is not the result of the lack of penetrance in single copy of EphA3 mutant, since it has been previously reported that duplicated maps are observed in only some single copy mutants.*

The lack of penetrance in the EphA3KI/+ mutants the reviewer is mentioning refers to recent data published by Owens et al., 2015, showing a single retino-collicular map in 25% of the EphA3KI/+ mutants, which is somewhat controversial. Previous, more precise, and detailed anatomical analyses of the retino-collicular map in the EphA3 single copy mutants showed consistent (in 100% of the EphA3KI/+ mutants) duplications (originally by Brown et al., 2000 and further demonstrated by Reber et al., 2004). In particular, projections from nasal RGCs to the caudal pole of the SC *ALWAYS* produced duplicated TZs as shown in Brown et al., 2000; Reber et al., 2004 and in the revised version (Figure 6—figure supplement 1).

*In addition to major concerns described above, there are also concerns in the presentation of the manuscript:*

*1) In some cases, images appear to be poorly selected to make the authors point. For example see Figure 2 which are described as being similar but are not of equal quality. Other examples include 3H-K, Figure 4, and Figure 6.*

We have added new images for Figure 2 and Figure 4 (single TZ panel in ephrin-A3KI/+). Panel 3 from Figure 6 was also changed for one of higher quality.

*2) In several cases proper labeling is missing or inappropriate on the figures. In Figure 1 labels are missing. In Figure 3 what are the arrows? (And the figures in Figure 3 are insufficient to claim normal eye-specific segregation in this mutant). It’s unclear what Figure 5 is amplifying? (And if its ephrinA3 why isn't it in WT retina?). More care should be paid to all figures and legends.*

Missing labels specifying the ganglion cell layer (GCL) are now presented in the revised version Figure 1.

Additional data obtained from the eye-specific segregation analyses are presented in Figure 3 of our revised version.

Figure 5 correspond to an amplicon of 90 bp located within the IRES sequence of the targeted Isl2 locus. This IRES sequence is present in the bi-cistronic RNA transcribed from the targeted Isl2 locus and is therefore present only in Isl2(+) RGCs in ephrin-A3KI/KI but not in SC or V1 nor in WT littermates (see Figure 5 legend).

*Reviewer #3:*

*The formation of topographic axonal maps is a key element of brain wiring. Visuotopic mapping is mainly instructed by the ephrin/Eph system, enabling forward, reverse, axon/target, axon/axon and cis signals. The actual contribution of each of these signals and their integration, has remained highly controversial. The authors specifically address the role of axonally located ephrin-As by a knock-in (ki) of ephrin-A3 into a random half-population of retinal ganglion cells (RGCs). Therefore, their paper is of major importance to the field. Their approach is valid, the experiments are thoroughly performed, carefully evaluated and lucidly presented in the paper. I have, however, two (somewhat) major issues that should be clarified before publication.*

*Based on our current understanding, a duplicated retinocollicular map in the KI/KI animals would be expected. Ki axons should grow to the posterior superior colliculus (SC) due to axon/target reverse signaling (Rashid T., et al. 2005). Wildtype axons should be anteriorly displaced due to axon/axon forward signaling (Suetterin P., et al., 2015). Intriguingly, the retinocollicular map is unaffected by the ki. In contrast, however, the V1-corticocollicular map is split up, notably indicating a role of ephrin-A3 on retinal axons in the chemotopic guidance of V1 cortical axons. The same projection has previously been claimed to be aligned by activity-dependent mechanisms (Triplett J., et al., 2009).*

Activity-dependent mechanisms have indeed been shown to align cortico-collicular mapping (Triplett et al., 2009), as mentioned by the reviewer. We suggest, as shown already for retino-collicular projections, that the mechanisms of cortico-collicular map alignment include *both* EphA/ephrin-A guidance cues *and* correlated activity (see subsection “Retinal EFNA3 in cortico-collicular mapping”, last paragraph).

*Although being very exciting, the new results are to some degree in conflict with previous observations, necessitating thorough control. Thus, I'm still not fully convinced, that an amount of KI ephrin-A3 arrives in the SC that could be functionally relevant to retinocollicular mapping. Maybe the pioneering retinocollicular mapping is more robust to changes than the following corticocollicular map. Immunostainings of isolated kiRGCs display labeling on the neurites (Figure 1'), but this is after 6 DIV and it remains unclear whether the shown neurites are axons or dendrites. The authors show that concomitant KI of EphA3 rescues the mapping effects of both knock-ins, but the mutual neutralization of ephrin-A3 and EphA3 might already occur in the retina. Therefore, the authors should provide some additional evidence for the axonal localization of the KI ephrin-a3 in the SC (e.g., Western blots, immunostainings on the SC or in-vitro assays).*

We chose to address the reviewer’s point by ephrin-A3 immunohistochemistry in WT and ephrin-A3KI mutant SCs (see response above, reviewer 1 §2).

As requested by the reviewer, we performed double-labelling of RGCs in vitro using ephrin-A3 and Tau (axonal marker) antibodies (see revised version Figure 1’), showing co-localization of both ephrin-A3 and Tau and suggesting the expression of ephrin-A3 on RGC axons in vitro (subsection “Knock-in mice for Efna3 ectopic expression in Isl2(+) RGCs show normal retino-collicular and 78 retino-geniculate projections”, first paragraph and Figure 1 legend).

Moreover, previous work in the laboratory demonstrated that the culture conditions used here do not allow the formation of RGC dendrites in vitro (see Steinmetz et al., 2006).

*My second major issue relates to the modeling. The presented model principally aids the understanding of the corticocollicular effects. In its present state, however, it has some deficiencies. First, either the key equation of the model, given in subsection “In silico replication of the duplication of the cortico-collicular map”, referring to two gradients on both, the target and the field of origin is wrong, or not all gradients used are specified.*

We used the equations of the model as described by Koulakov and Tsignakov, 2004. The key equation described refers to ONE EphA receptors gradient (R_A_) and its value at two locations in the retina/V1 (i and r’) and ONE ephrin-A ligands gradient (LA) and its value at two locations in the SC (i and r’), which is indeed confusing. In our revised version, the two locations in the retina/V1 are (i) and (j) and in the SC are (i’) and (j’) (see subsection “In silico replication of the duplication of the cortico-collicular map”).

*Second, I can't see how the fitting equations given for the ephrin gradients relate to the actual measurements.*

We thank the reviewer for pointing out our mistake (mis-copy). The equations indeed do not relate to the actual measurements of Figure 7. The equations computed in our model correspond to the retinal ephrin-As gradient transposed in the SC: during the retino-collicular projections, the x axis in the retina [nasal (0) – temporal (100)] is flipped onto the rostral (0) – caudal (100) axis in the SC where nasal RGCs project caudally and temporal RGCs project rostrally. We corrected the equations (subsection “In silico modelling and theoretical analysis of cortico-collicular map alignment”) and made this point clearer in our revised version.

*And third, for modeling corticocollicular mapping, the authors arbitrarily reduce the ephrin-A5 gradient to 80%. This is neither well motivated, nor has it been evaluated, how this affects the outcome.*

Different simulations were performed using proportion of retinal ephrin-A5 ranging from 0% to 100% (see table below). The 3-step model simulates the cortico-collicular map duplications observed experimentally in ephrin-A3KI mutants at 80% of retinal ephrin-A5 and higher (in bold blue in the table). Consistent with the literature showing a role of retinal

ephrin-A5 in RGCs fiber-fiber interactions (Suetterlin and Drescher, 2014), we chose to retain 80% of retinal ephrin-A5 in the simulations.

% CC duplication ( ± 95CI)% ephrin-A5ephrin-A3KI/+ephrinA3KI/KI5058 ( ± 2)64 ( ± 1.2)6052 ( ± 2.5)63 ( ± 1.7)7049 ( ± 1.7)52 ( ± 1.7)8041 ( ± 2.4)46 ( ± 1.2)9040 ( ± 2.5)48 ( ± 2.2)10041 ( ± 2.6)49 ( ± 2)experimental4347

[Editors’ note: the author responses to the re-review follow.]

*Essential revisions:*

*1) Confirm antibody specificity by staining WT vs. ephrin-A3 KO retinal and SC tissue, if possible testing tissue from the same litter (or source/background). This would validate the results, especially for the expression in the retina and optic nerve.*

As requested, we confirmed that both of our antibodies used to detect EFNA3 (Ab LSBio ref# LSC6547 and Ab Invitrogen ref# 36-7500) are specific by comparing the staining observed on P8 EfnA3-/- (n = 4 animals) versus P8 Efna3+/- (n = 2 animals) retina, optic nerve and SC sections (obtained from Efna3+/- and Efna3-/- heads kindly provided by Dr. D. Feldheim, UC Santa Cruz). As shown below, both antibodies show no specific staining in Efna3-/- sections, validating their specificity.

Author response image 5.Immunohistochemistry on P8 Efna3-/- (A, B, C, G, H, I) and Efna3+/- (D, E, F, J, K, L) tissue sections using EFNA3 antibodies from LSBio (LSC6547) (A-F) and Invitrogen (36-7500) (G-L).Scale bars, 100µm (A, C, D, F, G, I, J, L), 50 µm (B, E, H, K).**DOI:**
http://dx.doi.org/10.7554/eLife.20470.027

Figures B and C have been added to the main manuscript, Figure 5—figure supplement 1 and Figure 4—figure supplement 4 (subsection “Cortico-collicular maps are duplicated in *Isl2(-)* Efna3KI mutants”, last paragraph) respectively.

*2) Include results of the expression as shown in the rebuttal in the manuscript itself, in Supplemental Data.*

As requested, these expression data now appear in the third version of our manuscript as Figure 4—figure supplement 3, Figure 4—figure supplement 4, and Figure 5—figure supplement 1 (legends and related text in subsection “Cortico-collicular maps are duplicated in Isl2-Efna3KI mutants”, last paragraph).

*3) More thoroughly address the "dissonance" in ephrin-A3 expression along RGC axons in the Discussion.*

This point has been discussed in our third version of the manuscript, in the last paragraph of the subsection “Retinal EFNA3 in cortico-collicular mapping”.

*4) Ephrin-A3 is present on axons, but below the detection limit; thus, there cannot be 100% transfer of the retinal ephrin-A3 into the axons. You should include this finding in your model.*

As mentioned by reviewer 3, axonal ephrin-A3 expression (in the distal part of the optic nerve as well as in the SC only) is below our immunohistochemical detection limit, most likely because of dissolution of ephrin-A3 along the axons. This dissolution is very likely to occur for all EphA receptor and ephrin-A ligands in this system (see Dutting et al., 1999; Triplett et al., 2009, Supplementary Figure 4). Therefore, if we reduce transfer, we should do it for all guidance cues, not just retinal ephrin-A3. However, given that we have no plausible way of setting realistic values for all these dissolution rates, we believe any adjustments would be too speculative and under-constrained to be useful.